# MYC competes with MiT/TFE in regulating lysosomal biogenesis and autophagy through an epigenetic rheostat

Ida Annunziata [1], Diantha van de Vlekkert [1], Elmar Wolf [2], David Finkelstein[3], Geoffrey Neale [4], Eda Machado [1], Rosario Mosca [1], Yvan Campos[1], Heather Tillman [5], Martine F. Roussel [6], Jason Andrew Weesner [1,7], Leigh Ellen Fremuth [1,7], Xiaohui Qiu [1], Min-Joon Han[8], Gerard C. Grosveld [1] & Alessandra d'Azzo [1]

Coordinated regulation of the lysosomal and autophagic systems ensures basal catabolism and normal cell physiology, and failure of either system causes disease. Here we describe an epigenetic rheostat orchestrated by c-MYC and histone deacetylases that inhibits lysosomal and autophagic biogenesis by concomitantly repressing the expression of the transcription factors MiT/TFE and FOXH1, and that of lysosomal and autophagy genes. Inhibition of histone deacetylases abates c-MYC binding to the promoters of lysosomal and autophagy genes, granting promoter occupancy to the MiT/TFE members, TFEB and TFE3, and/or the autophagy regulator FOXH1. In pluripotent stem cells and cancer, suppression of lysosomal and autophagic function is directly downstream of *c-MYC* overexpression and may represent a hallmark of malignant transformation. We propose that, by determining the fate of these catabolic systems, this hierarchical switch regulates the adaptive response of cells to pathological and physiological cues that could be exploited therapeutically.

[1] Department of Genetics, St. Jude Children's Research Hospital, Memphis, TN 38105, USA. [2] Department of Biochemistry and Molecular Biology, Biocenter, University of Würzburg, Würzburg 97074, Germany. [3] Department of Computational Biology, St. Jude Children's Research Hospital, Memphis, TN 38105, USA. [4] Hartwell Center, St. Jude Children's Research Hospital, Memphis, TN 38105, USA. [5] Department of Pathology, St. Jude Children's Research Hospital, Memphis, TN 38105, USA. [6] Department of Tumor Cell Biology, St. Jude Children's Research Hospital, Memphis, TN 38105, USA. [7] Department of Anatomy and Neurobiology, College of Graduate Health Sciences, University of Tennessee Health Science Center, Memphis, TN 38163, USA. [8] Department of Hematology, St. Jude Children's Research Hospital, Memphis, TN 38105, USA. Correspondence and requests for materials should be addressed to A.d. (email: sandra.dazzo@stjude.org)

Eukaryotic development and differentiation programs depend on the fine-tuned regulation of gene expression, which is achieved by continuous remodeling/modification of chromatin through reversible epigenetic marks on histone tails[1]. The latter modulates the assembly/compaction of chromatin and the recruitment of transcription factors[2]. Among the modifications that change chromatin structure, histone acetylation or deacetylation of specific lysine residues by histone acetylases or deacetylases (HDACs) allows for the rapid adaptation of cells to extra- and intracellular signals[3]. In mammals, HDACs comprise a family of 18 genes, which are grouped, based on their homology to their yeast counterparts, into classes I to IV. Classes I, II, and IV include 11 $Zn^{2+}$-dependent family members, which are referred to as "classical" HDACs[4]. In general, HDACs display limited substrate selectivity and rely on the association with transcription factors or repressor complexes to attain their specificity at target DNA sites[5].

Chromatin modifications by HDACs influence a multitude of cellular pathways through repression of transcription of metabolic genes. Overexpressed or deregulated HDACs have been linked to human conditions mostly associated with aging, such as cancer, diabetes, cardiac hypertrophy, and neurodegenerative diseases[6–8]. Inhibition of HDACs promotes growth arrest and cell differentiation or apoptosis[9]. For this reason, the Food and Drug Administration (FDA) has approved HDAC inhibitors for the treatment of various malignancies, in which they cause altered expression of only a small subset of genes, suggesting that acetylation/deacetylation is restricted to specific chromatin regions[10]. HDAC inhibitors have also been used to reverse disease-associated epigenetic states in adult cardiovascular, neurodegenerative and inflammatory diseases, and more recently in some of the pediatric lysosomal storage diseases (LSDs)[11–14].

Lysosomes control the breakdown, processing or recycling of long-lived proteins, nucleic acids, carbohydrates, and complex lipids that reach the organelles through the biosynthetic, endocytic, phagocytic or autophagic route[15]. These processes are carried out by a large range of hydrolytic enzymes, membrane proteins, transporters, and ion channels that are ubiquitously but differentially expressed in different cell types and tissues. The importance of a fully functional lysosomal system in maintaining cell homeostasis is evidenced in the complex pathobiology of LSDs. Endo-lysosomal accumulation of undigested or partially digested substrates in cells of virtually all organs is the hallmark of these diseases and determines phenotypic penetrance and outcome. The lysosomal system is strictly connected to the autophagic system because autophagosomes need to fuse with lysosomes to execute the degradative stage of the pathway[16]. Therefore, impaired lysosomal activity leads also to the accumulation of autophagic substrates and autophagic disfunction[17]. Autophagy is a highly conserved catabolic process mainly activated during starvation, allowing cells to generate energy during nutrient deprivation.

Both the lysosomal and autophagic systems have recently gained attention as modulators of nutrient sensing and signaling[17,18]. The constituents of this organellar network are transcriptionally regulated by members of the microphthalmia-associated family of basic helix loop helix (b-HLH) leucine zipper transcription factors (MiT/TFE), which include TFEB, TFEC, TFE3, and MITF[19,20]. When altered, MiT/TFE members induce cancer and other severe diseases[21]. All MiT/TFE transcription factors recognize a unique enhancer box (E-box) DNA motif [also referred to as CLEAR (coordinated lysosomal expression and regulation)] within the proximal promoters of lysosomal and autophagy genes[20,22,23], and regulate cellular catabolism and nutrient-dependent lysosomal response[19,24].

Another prototypical member of the b-HLH leucine zipper class of transcription factors is c-MYC (hereafter referred to as MYC), which also binds to E-boxes near the core promoter elements of target genes[25,26]. MYC functions as master regulator of cellular metabolism and proliferation[27]. Under physiological conditions, MYC transcription is controlled by developmental and mitogenic cues, and decreases in differentiated cells in absence of additional activating signals. The MYC protein is short lived in proliferating cells, which adds an extra checkpoint to avoid transformation. In addition to promoting cell growth and proliferation, MYC inhibits terminal differentiation of cells via activation or repression of target genes. Both MYC functions depend on the recruitment of chromatin-modifying complexes, including HDACs, which, by changing accessibility to the transcriptional machinery, allow or prevent MYC-driven transcription of the corresponding genes[28]. Under physiological conditions this mechanism is strictly regulated, but in cancer cells constitutively elevated expression of MYC shifts the balance toward abnormal activation or repression of MYC responsive genes[27,29]. When overexpressed, MYC is a potent oncogene, and its constitutive high-expression drives many tumor types and is often associated with cancer aggressiveness and poor prognosis[29].

Here we define a previously unknown antagonistic role between HDAC/MYC and MiT/TFE in the epigenetic and transcriptional control of the two-major cellular catabolic machineries. We further unveil that inhibition of HDACs and reduction of MYC levels displace MYC from E-Box promoter sites of lysosomal and autophagy genes, which grants occupancy of the same sites to MIT/TFE members and results in activation of lysosomal biogenesis and autophagy. Finally, we show that this competing transcriptional rheostat is established during cell reprogramming and differentiation in both normal and disease conditions.

## Results

**HDACs regulate lysosomal biogenesis and MiT/TFE members.** To investigate whether epigenetic marks regulate lysosomal function, we tested whether suppression of histone acetylation with suberoylanilide hydroxamic acid (SAHA)[30], an FDA approved HDAC pan-inhibitor, altered the expression of lysosomal genes. Microarray expression profiles, obtained from either DMSO- or SAHA-treated HeLa cells, identified differentially activated or suppressed pathways by gene set enrichment analysis (GSEA) and database for annotation, visualization, and integrated discovery (DAVID) analysis (Fig. 1a, Supplementary Fig. 1a, and Supplementary Table 1 and Supplementary Data 1).

Activation of the lysosomal pathway was among the top-ranked responses by GSEA and DAVID in SAHA-treated cells (Fig. 1a and Supplementary Table 1 and Supplementary Data 1), while the most suppressed pathways were related, as previously reported[31], to cell cycle and mitotic division (Supplementary Data 1). HDAC inhibition in HeLa cells increased the mRNA abundance of 19 out of 22 lysosomal genes tested (Fig. 1b); among them the sialidase-encoding gene neuraminidase 1 (NEU1) was the top responder (Fig. 1b). To assess the generality of these findings, we also tested the effects of SAHA treatment in additional human cancer cell lines (RH30, rhabdomyosarcoma and Sy5y, neuroblastoma), and in primary skin fibroblasts. The results were comparable to those obtained with HeLa cells (Supplementary Fig. 1b–d). In addition to the data presented here, we also confirmed activation of global lysosomal gene expression by querying publicly available datasets obtained from endothelial-, transformed lymphoblastoid- and breast cancer-cells treated with SAHA (Supplementary Fig. 1e–g). The strong

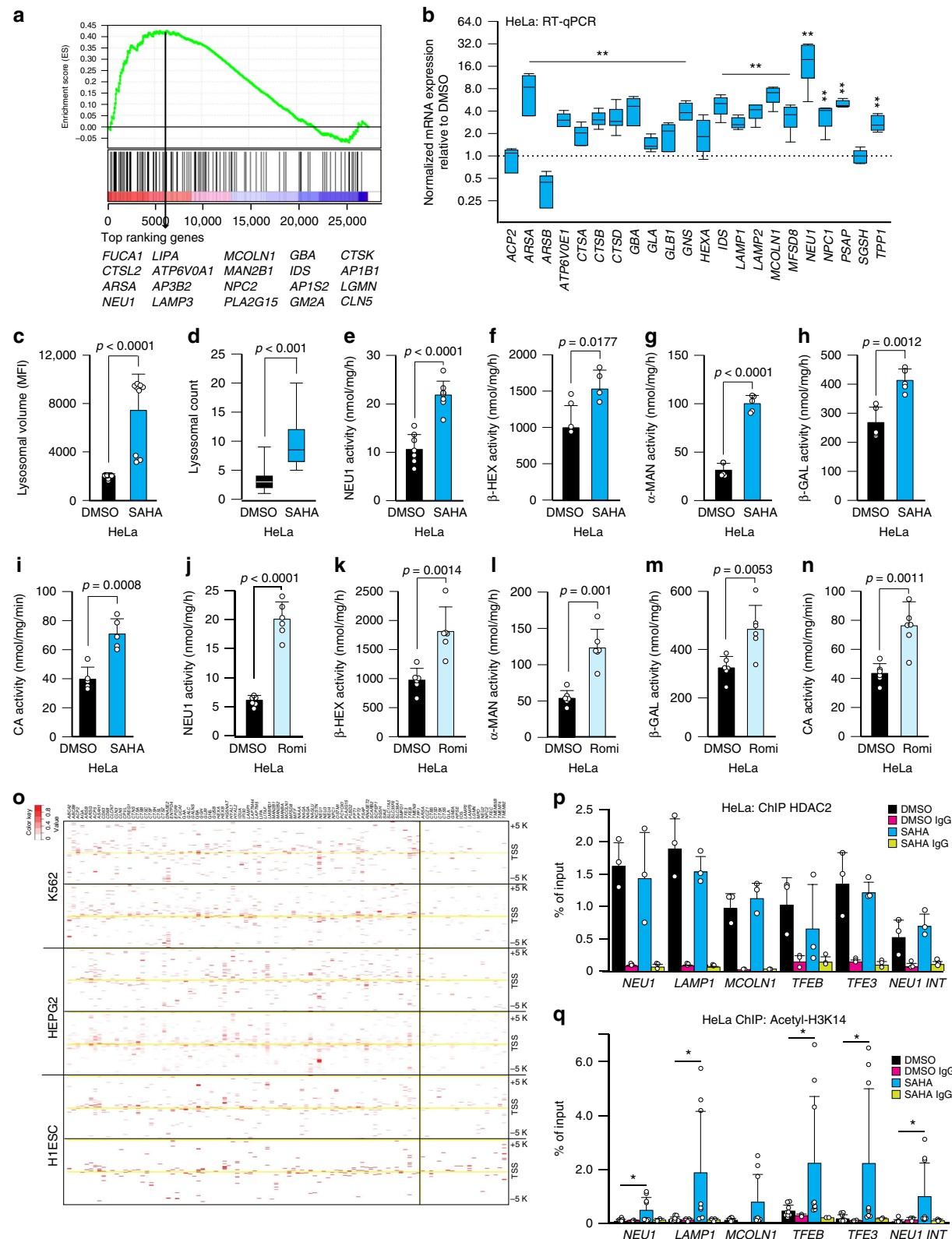

induction of lysosomal biogenesis was further proven by the expansion of the lysosomal compartment (Fig. 1c, d), detected by flow cytometry of lysotracker green positive lysosomes, the increased levels of various membrane and soluble lysosomal components (Supplementary Fig. 2a–f), and the enhanced enzymatic activity of glycosidases, including NEU1, and cathepsins (Fig. 1e–i and Supplementary Fig. 2g–i). While further

details are beyond the scope of the present study, the difference between NEU1 mRNA/protein levels and NEU1 enzymatic activity observed upon SAHA treatment could be explained by the low-specific activity of NEU1 towards the synthetic substrate and/or by the rate limiting amount of PPCA (Protective Protein/Cathepsin A) available for chaperoning and activating NEU1 in lysosomes[32].

**Fig. 1** HDACs epigenetically regulate the expression of lysosomal genes. **a** GSEA demonstrated significant activation of the lysosomal pathway in HeLa cells treated with SAHA (20 μM for 24 h). The top 20 upregulated genes are shown at the bottom of the plot. **b** Expression analysis of lysosomal genes after SAHA treatment (20 μM for 24 h). The box and whisker plot show normalized expression of the lysosomal mRNAs in SAHA-treated HeLa cells relative to that in DMSO-treated cells. $n = 5$ biologically independent samples. **c** Lysosomal volume was measured by FACS analysis as mean fluorescence intensity (MFI) after lysotracker staining of HeLa cells treated with SAHA (20 μM for 24 h) and compared to cells treated with DMSO, $n = 9$ biologically independent samples. **d** Lysosomal count of transmission electron microscopy images $n = 20$ images. **e–n** Activity assays for NEU1 ($n = 7$ biologically independent samples), β-hexosaminidase (β-HEX) ($n = 4$ biologically independent samples), α-mannosidase (α-MAN) ($n = 5$ biologically independent samples), β-galactosidase (β-GAL) ($n = 5$ biologically independent samples) and cathepsin A (CA) ($n = 5$ biologically independent samples) in HeLa cells treated with DMSO or SAHA (**e–i**) (20 μM for 24 h), or in HeLa cells treated with DMSO or romidepsin (**j-n**) (romidepsin, Romi, 10 nM for 24 h; $n = 6$ biologically independent samples). **o** HDAC2 occupancy was analyzed using ChIP-seq datasets available at the ENCODE/Haib project. Genes shown on the left of the vertical black line are occupied by HDAC2. $p = 4.65e-11$, odds ratio = 4.4, Fisher's exact test. **p** ChIP analysis of the promoters of NEU1, LAMP1, MCOLN1, TFEB, and TFE3 using anti-HDAC2 antibody in HeLa cells treated with SAHA (20 μM for 24 h) or DMSO ($n = 3$ biologically independent samples). Oligos encompassing a genomic region at +5 Kb from the TSS of NEU1 (NEU1 INT) were used as control for non-specic antibody binding ($n = 3$ biologically independent samples); IgG control $n = 3$ biologically independent samples. **q** ChIP analysis of the promoters of NEU1, LAMP1, MCOLN1, TFEB, and TFE3 and NEU1 INT using acetyl histone H3 Lys 14 antibody (Acetyl-H3K14) in HeLa cells treated with SAHA (20 μM for 24 h) or DMSO ($n = 8$ biologically independent samples); IgG control $n = 3$ biologically independent samples. Boxes represent the mean value and bar inside the box represents median value; upper bar represents maximum of distribution; lower bar represents minimum of distribution (95% confidence level). Graphs shown in (**c–n**) and (**p, q**) are presented as mean ±SD. Statistical analysis was performed using Student $t$-test

In order to narrow the pan effect of SAHA inhibition on HDACs, we also tested in the same experimental setup the effects of romidepsin, a more potent and selective class I HDAC inhibitor[33]. The activation of lysosomal gene expression (Supplementary Fig. 3a) and consequent increase in the activity of several lysosomal enzymes (Fig. 1j–n) closely recapitulated the results with SAHA, strongly supporting the idea that members of class I HDACs are primarily responsible for the epigenetic suppression of lysosomal biogenesis.

We next investigated whether HDACs engage the promoters of lysosomal genes by querying publicly available ChIP datasets for the class I HDACs. For this analysis we used HDAC2 as a proxy (http://genome.ucsc.edu/cgi-bin/hgTrackUi?db=hg19&g=wgEncodeHaibTfbs), as it is one of the most widely expressed nuclear HDACs[34]. We found that most (82 of 102) lysosomal genes were overrepresented among the HDAC2-bound targets (Fig. 1o). These results were confirmed by ChIP performed with HeLa cells treated with either SAHA or DMSO using an anti-HDAC2 antibody (Fig. 1p). This assay also revealed that HDAC2 occupied the promoters of both TFEB and TFE3 (Fig. 1p), two of the MiT/TFE members known to regulate lysosomal function and metabolism[20,22]. It is important to notice that inhibition of HDAC2 with SAHA did not alter its binding capacity to the promoters; this is because SAHA specifically affects the histone deacetylase activity of HDACs without altering their protein levels[35]. Remarkably, silencing of only HDAC2 (Supplementary Fig. 3b, c) was sufficient to increase the activity of lysosomal enzymes (Supplementary Fig. 3d–g) in a manner comparable to that obtained upon HDAC inhibition. Activation of gene transcription by inhibiting HDACs was also measured by increased acetylation of histone 3 (H3) on lysine 14 (H3K14) of the promoter regions of several lysosomal genes as well as of TFEB and TFE3 genes (Fig. 1q). Together these results indicate that HDACs, and specifically HDAC2, epigenetically control the expression levels not only of a plethora of lysosomal genes but also of the MiT/TFE transcription factors.

**MYC represses lysosomal biogenesis.** In search for putative transcription factor binding sites in the promoters of lysosomal genes bound by HDAC2, we performed motif analysis and identified the E-box as the motif with the highest probability of occupancy. E-box binding sites are recognized by the b-HLH family of transcription factors (Fig. 2a) that include MiT/TFE members and MYC, the master regulator of metabolism[27], The potential engagement of MYC at lysosomal gene promoters was particularly intriguing because it has been well documented that MYC transcription and protein levels are directly modulated by HDAC activity[28,36,37] and that MYC and HDACs interact[38,39]. In line with these observations we showed that silencing of HDAC2 drastically reduced MYC protein levels (Fig. 2b, c and Supplementary Fig. 4a, b), that MYC and HDAC2 co-immunoprecipitated (Fig. 2d, e and Supplementary Fig. 2c, d) and that HDAC2 was bound to the MYC promoter (Fig. 2f). We noticed that the E-box motif recognized by MYC[25] remarkably overlaps with the CLEAR motif recognized by TFEB and TFE3, raising the possibility that MYC binds the promoters of lysosomal genes. To test this hypothesis, we queried ChIP-seq datasets performed with anti-MYC antibody[29,40] and found that MYC occupied not only the promoters of lysosomal genes (Fig. 2g, h and Supplementary Table 2 and Supplementary Data 2) but also those of MiT/TFE family members TFEB and TFE3 (Fig. 2i and Supplementary Fig. 4e, Supplementary Data 2 and Supplementary Table 3). In addition, ChIP analyses of HeLa cells, treated or not with SAHA, confirmed that in untreated cells MYC occupied the promoters of TFEB and TFE3, but was displaced upon inhibition of HDACs (Fig. 2j). We also observed that MYC was bound to its own promoter in control cells, but this binding was reduced in SAHA-treated cells (Fig. 2k). Furthermore, chromatin precipitated with the MYC antibody and sequentially precipitated with HDAC2 antibody revealed that TFEB and TFE3 promoters were co-occupied by MYC and HDAC2 (Fig. 2l).

In agreement with these observations, MYC mRNA and protein levels were significantly downregulated upon treatment with HDAC inhibitors (Fig. 3a, b and Supplementary Fig. 4a, b and Supplementary Fig. 4f–h). In contrast, the expression of the MiT/TFE members was significantly increased upon SAHA/romidepsin treatment, albeit in a cell-specific manner, which is likely due to the relative abundance of these transcription factors in different cell types: TFE3 was increased in HeLa cells (Fig. 3a, b), MITF, TFEB, and TFE3 were all increased in RH30 (Supplementary Fig. 4f) and Sy5y (Supplementary Fig. 4g) cell lines, while MITF was increased exclusively in skin primary fibroblasts (Supplementary Fig. 4h). Performing ChIP assays of HeLa cells treated with SAHA, we further demonstrated that MYC downregulation enabled binding of TFEB and TFE3 to the promoters of lysosomal genes and also to their own respective promoters (Fig. 3c–e). This mechanism may allow for robust but tightly controlled activation of lysosomal gene expression, as consequence of acetylation of histones.

To unequivocally demonstrate that TFEB and TFE3 are the primary drivers of lysosomal biogenesis under SAHA treatment,

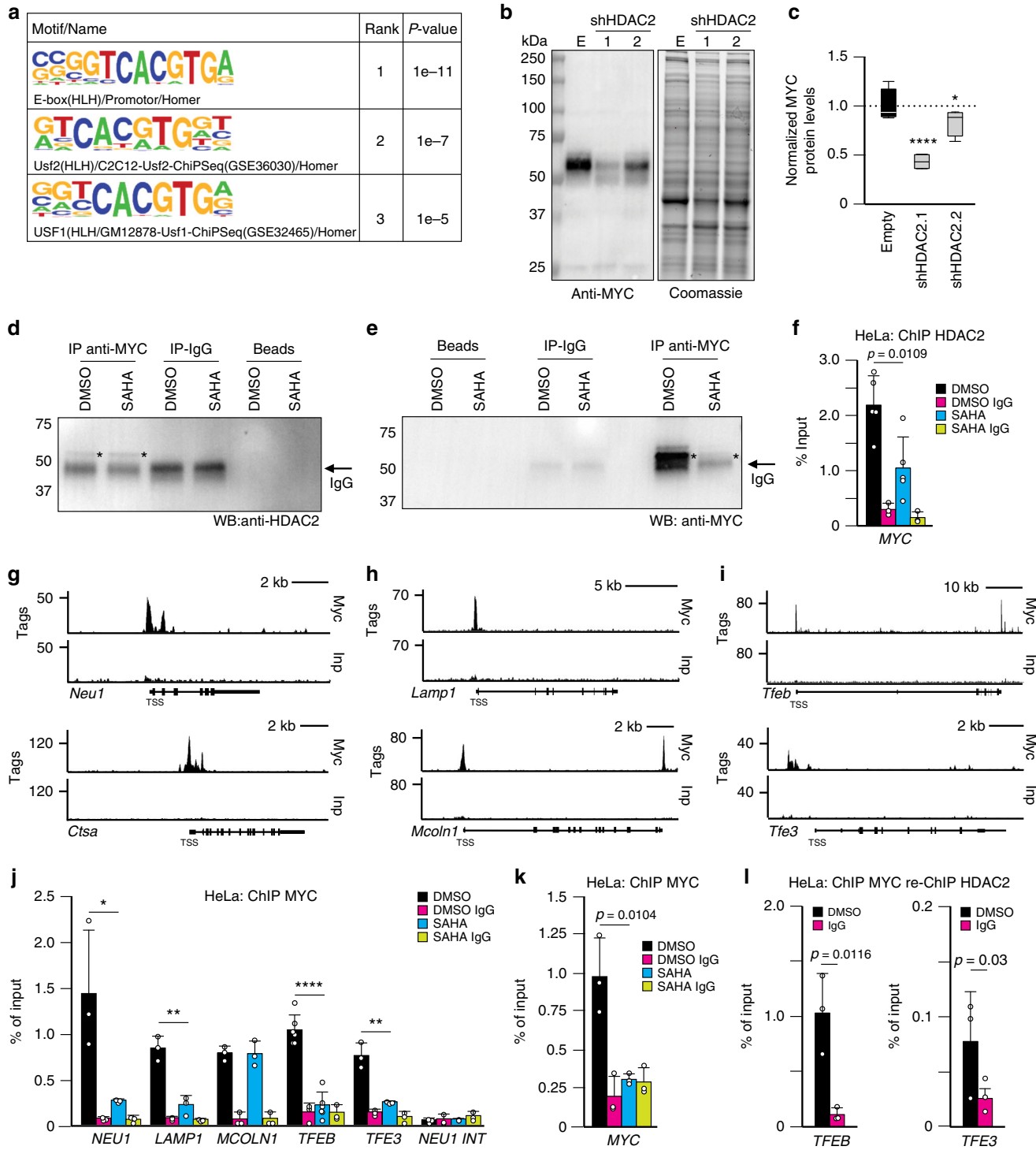

we tested the effects of this inhibitor in murine embryonic fibroblasts (MEFs) that had CRISPR-mediated ablation of both transcription factors (dKO)[41] (Supplementary Fig. 4i, j). In spite of the complete absence of these transcription factors, we still measured activation of lysosomal biogenesis upon SAHA treatment of these dKO cells (Fig. 3f, g). We found that these unexpected results were likely due to the upregulation in the dKO cells of Mitf expression (Supplementary Fig. 4k). This finding is in agreement with the notion that members of the Mit/Tfe family function cooperatively or redundantly in the regulation of lysosomal biogenesis[42,43]. Upon shRNA silencing of Mitf in the dKO cells, followed by SAHA treatment (Supplementary Fig. 4k),

we observed a marked reduction in lysosomal gene expression (Fig. 3f, g) and reduced NEU1 activity (Fig. 3h) compared to WT MEFs. Together these results confirm that MIT/TFE members are indeed responsible for the activation of lysosomal genes after HDAC inhibition.

To further understand how MYC fine-tunes lysosomal biogenesis, we tested a set of mouse group 3 medulloblastoma cell lines overexpressing Myc[40,44]. In those cells, the expression of murine Mit/Tfe members was significantly downregulated, as was that of all tested lysosomal genes (Fig. 4a, b and Supplementary Data 2). The latter was accompanied by a reduction of the lysosomal volume (Supplementary Fig. 4l), decreased lysosomal

**Fig. 2** MYC occupies the promoters of lysosomal genes and that of TFEB and TFE3. **a** Motif analysis using HDAC2-binding sites present in lysosomal genes. **b** Left, silencing of HDAC2 downregulated MYC protein expression in HeLa cells. Right, Coomassie stained immunoblot used as the loading control. **c** Quantification of MYC levels in HDAC2 silenced HeLa cells normalized to loading control ($n = 3$ independent experiments). **d** Co-immunoprecipitation of MYC with HDAC2 from lysates obtained from HeLa cells. HDAC2 band is labeled with an asterisk. Heavy chain IgG is labeled with an arrow. **e** Immunoprecipitation of MYC followed by immunoblot of MYC protein. MYC band is labeled with an asterisk. Heavy chain IgG is labeled with an arrow. **f** Graph represents HDAC2 binding to the promoter of *MYC* ($n = 5$ biologically independent samples) in HeLa cells treated with DMSO and SAHA (20 μM for 24 h); IgG control ($n = 3$ biologically independent samples). **g–i** Myc binding to the promoters of (**g**) *Neu1, Ctsa*, (**h**) *Lamp1, Mcoln1* and **i** *Tfeb and Tfe3* was analyzed in ChIP-seq datasets performed with anti-Myc antibody in mouse group 3 medulloblastoma cells overexpressing *Myc* (*Trp53*$^{-/-}$ overexpressing *Myc*). Input DNA (Inp) serves as reference. **j**, **k** Histograms represent MYC binding to the promoters of (**j**) *NEU1* ($n = 3$ biologically independent samples), *LAMP1* ($n = 3$ biologically independent samples), *MCOLN1* ($n = 3$ biologically independent samples), *TFEB* ($n = 6$ biologically independent samples), *TFE3* ($n = 3$ biologically independent samples) and (**k**) *MYC* ($n = 3$ biologically independent samples) in HeLa cells treated with SAHA (20 μM for 24 h) or DMSO. *NEU1 INT* oligos were used as negative control for non-specific antibody binding ($n = 3$ biologically independent samples); IgG control ($n = 3$ biologically independent samples). **l** Sequential ChIP experiments were performed from HeLa cells with anti-MYC antibody followed by anti-HDAC2 antibody and analyzed by RT-qPCR at the promoter region of *TFEB* and *TFE3* ($n = 3$ biologically independent samples); IgG control ($n = 3$ biologically independent samples). All the graphs are presented as mean ± SD. Statistical analysis was performed using the Student *t*-test. *$p < 0.05$, **$p < 0.01$, ****$p < 0.0001$

Neu1 enzyme activity and decreased Lamp1 protein levels (Supplementary Fig. 4m–o). In contrast, computational analysis of U2OS osteosarcoma cells silenced for *MYC*[29], showed a transcriptional activation of the *MIT/TFE* members and of lysosomal genes (Fig. 4c, d). To corroborate these findings, we tested lysosomal gene expression in HeLa cells silenced for MYC (Supplementary Fig 4p, q). We demonstrated that merely silencing MYC already induced the expression of lysosomal genes that was further enhanced upon HDAC inhibition (Fig. 4e). This resulted in increased enzymatic activity of NEU1 (Fig. 4f).

**The HDAC-MYC repressor rheostat regulates autophagy.** Considering the high degree of crosstalk between the lysosomal and the autophagic systems, we explored whether autophagy is also controlled by the HDAC/MYC axis. Inhibition of HDACs significantly increased the expression of many autophagy genes in both primary fibroblasts and cancer cell lines (Fig. 5a, Supplementary Fig. 5a–e). These genes encode proteins that orchestrate the induction, formation and maturation of autophagosomes and the selection, expansion and degradation of their cargo[45]. Upon SAHA treatment, analysis of the lipidated LC3 protein and the LC3II/LC3I ratios (Supplementary Fig. 5f–k), which specify autophagosomes[46], confirmed the induction of autophagy. In addition, the appearance of GFP-LC3B puncta (Fig. 5b, c) and the increase in autophagic flux, assessed in cells treated or not with the inhibitor bafilomycin A1, (Fig. 5d, e and Supplementary Fig. 5l), proved the occurrence in SAHA-treated cells of autophagosome synthesis and delivery of autophagic substrates to the lysosome for degradation.

We next queried the ENCODE/Haib dataset for engagement of HDAC2 with the promoters of autophagy genes. Most (45 of 53) of the autophagy genes analyzed were indeed occupied by HDAC2 (Fig. 5f). ChIP assays of chromatin isolated from HeLa cells with anti-HDAC2 antibody showed that HDAC2 was bound to the promoter of the autophagy gene *MAP1LC3B* (Fig. 5g). In contrast, inhibition of HDAC activity by treatment with SAHA led to increased acetylation of H3K14 histone mark on the *MAP1LC3B* promoter region (Fig. 5h). Notably, transcription of autophagy genes was reduced upon SAHA treatment in MEFs dKO for Tfeb and Tfe3 and silenced for Mitf, suggesting that these transcription factors cooperate in the induction of autophagy upon SAHA treatment (Fig. 5i, j).

To further test the link between autophagy and MYC activation, we also interrogated Myc-ChIP-seq binding datasets[29,40] from group 3 mouse medulloblastoma and found that the promoters of several autophagy genes were indeed occupied by Myc (Fig. 6a and Supplementary Data 3). These

results were again confirmed by ChIP assays with anti-MYC antibody showing the binding of MYC to the promoter of the *MAP1LC3B* gene in SAHA-treated HeLa cells (Fig. 6b). These observations suggest a model in which MYC activation represses autophagy genes. In fact, this is the case because medulloblastoma cells overexpressing *Myc*[40] showed downregulated expression of autophagy genes (Fig. 6c), resulting in a decreased LC3II/LC3I ratio (Fig. 6d, e and Supplementary Fig. 6a). In contrast, querying the U2OS osteosarcoma dataset[29] showed enhanced transcription of several autophagy genes (Fig. 6f). Consistent with these results, silencing of MYC in HeLa cells led to a statistically significant increase in the expression of several autophagy genes, which was further enhanced by SAHA treatment (Fig. 6g).

Altogether, these data underscore that HDACs and MYC cooperate in suppressing not only lysosomal biogenesis but also autophagy.

**FOXH1 regulates autophagy.** To uncover the transcriptional machinery cooperating with HDACs in the regulation of autophagy, we performed motif-discovery analysis and identified the TGT[GT][GT]ATT motif, which is bound by the FOXH1 transcription factor (Fig. 7a). FOXH1 was first described as a developmental gene, because it induces mesoderm specification in cooperation with activin, a member of the TGF-β family. In mammals, FOXH1 also mediates TGF-β-type signaling[47].

To investigate if autophagy was induced by FOXH1, we overexpressed it in HeLa cells (Supplementary Fig. 6b) and showed that the mRNA levels of many autophagy genes were significantly increased (Fig. 7b). This was paralleled by activation of autophagy, as measured by the lipidation of LC3B (Fig. 7c, d and Supplementary Fig. 6c). We next questioned whether HDACs modulate the levels of FOXH1. We demonstrated that SAHA treatment of HeLa, RH30 and Sy5y cells, but not of primary human fibroblasts, induced *FOXH1* mRNA expression (Fig. 7e and Supplementary Fig. 6d–f), suggesting a cell type-specific activation of FOXH1 in response to SAHA. Enhanced *FOXH1* transcription was accompanied by the increase of FOXH1 protein in HeLa cells treated with SAHA (Fig. 7f, g). In addition, by querying MYC-ChIP datasets derived from U2OS cells with induced MYC expression[29], we confirmed MYC binding to the promoter of *FOXH1*, which resulted in reduced expression of *FOXH1* (Fig. 7h, i). Similarly, overexpression of *Myc* in mouse medulloblastoma cells negatively regulated *Foxh1* expression (Fig. 7j). In contrast, silencing of MYC in U2OS cells[29] resulted in the enhanced transcription of *FOXH1* (Fig. 7k). Based on these results, we propose that FOXH1 is an additional activator of

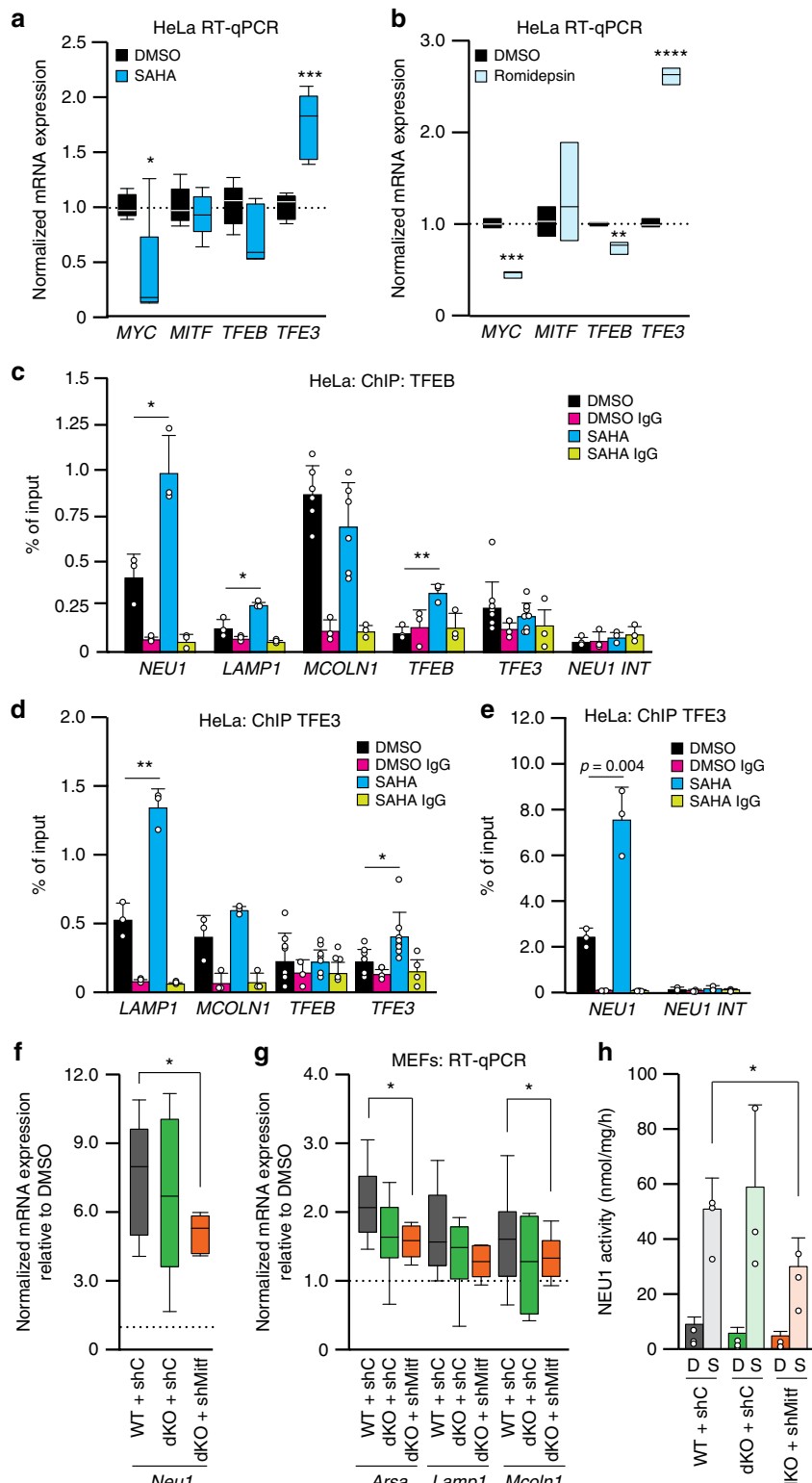

autophagy, specifically regulated by HDAC and MYC that may control this process in a temporal and cell/tissue-specific manner.

**Functional effects of HDAC inhibition in the LSD sialidosis.** Given that HDAC inhibitors were shown to reverse some of the phenotypes in the LSDs Niemann Pick type C and Gaucher disease[11,13,48], we tested the potential therapeutic effects of HDAC inhibition in fibroblasts derived from patients with

sialidosis, a rare neurosomatic LSD caused by mutations in the NEU1 gene[49]. Sialidosis is an orphan disorder for which there is currently no cure[49]. Sialidosis patients are usually classified based on the age of onset, severity of the symptoms and levels of NEU1 residual activity in type I (less severe) and type II (more severe)[49]. Patient-derived fibroblasts treated with SAHA showed increased levels of their mutant NEU1 mRNA, albeit to a different extent (Fig. 8a). Immunoblots probed with anti-NEU1 antibody revealed a marked increase of the mutant

**Fig. 3** MYC antagonizes MiT/TFE members. **a**, **b** RT-qPCR of *MYC*, *MITF*, *TFEB*, and *TFE3* in (**a**) SAHA-treated (20μM for 24 h; n = 5 biologically independent samples) or **b** romidepsin-treated (10 nM for 24 h) HeLa cells (n = 3 biologically independent samples) and compared to DMSO-treated cells. **c** ChIP analyses of *NEU1* (n = 3 biologically independent samples), *LAMP1* (n = 3 biologically independent samples), *MCOLN1* (n = 6 biologically independent samples), *TFEB* (n = 3 biologically independent samples), and *TFE3* (n = 9 biologically independent samples) promoters using anti-TFEB antibody in HeLa cells treated with SAHA (20 μM for 24 h) or DMSO. *NEU1 INT* oligos were used as negative control for non-specific antibody binding (n = 3 biologically independent samples); IgG control (n = 3 biologically independent samples). **d**, **e** ChIP analyses of the promoters of (**d**) *LAMP1* (n = 3 biologically independent samples), *MCOLN1* (n = 3 biologically independent samples), *TFEB* (n = 7 biologically independent samples), *TFE3* (n = 6 biologically independent samples) and (**e**) *NEU1* (n = 3 biologically independent samples) using an anti-TFE3 antibody in HeLa cells treated with SAHA (20 μM for 24 h) or DMSO. *NEU1 INT* oligos were used as negative control for non-specific antibody binding (n = 3 biologically independent samples); IgG control (n = 3 biologically independent samples). **f**, **g** RT-qPCR of (**f**) *Neu1* (**g**) *Arsa*, *Lamp1*, and *Mcoln1* was performed in mouse embryonic fibroblasts (MEFs), in which *Tfeb* and *Tfe3* were knocked out via CRISPR-technology (dKO) and in dKO with *Mitf* silencing (dKO + shMitf) (n ≥ 6 biologically independent samples) and treated with SAHA (20μM for 24 h) or DMSO. shC refers to cells transduced with shRNA control lentivirus. Statistical analysis was performed comparing the normalized expression of the lysosomal genes in MEFs WT + shC versus dKO + shC and MEFs WT + shC versus dKO + shMitf. **h** Activity assay for NEU1, in MEFs WT, *Tfeb* and *Tfe3* dKO transduced with a sh lentivirus control (ShC) or with a lentiviral vector targeting *Mitf* (dKO + shMitf) (n = 3 biologically independent samples). MEFs of different genotypes were treated with DMSO and SAHA (20 μM for 24 h). Statistical analysis was performed comparing the activity of NEU1 after SAHA treatment in MEFs WT + shC to dKO + shC and dKO + shMitf. Boxes represent the mean value and bar inside the box represents median value; upper bar represents maximum of distribution; lower bar represents minimum of distribution (95% confidence level). Graphs are presented as mean ± SD. Statistical analysis was performed using the Student *t*-test. *p < 0.05, **p < 0.01, ****p < 0.0001

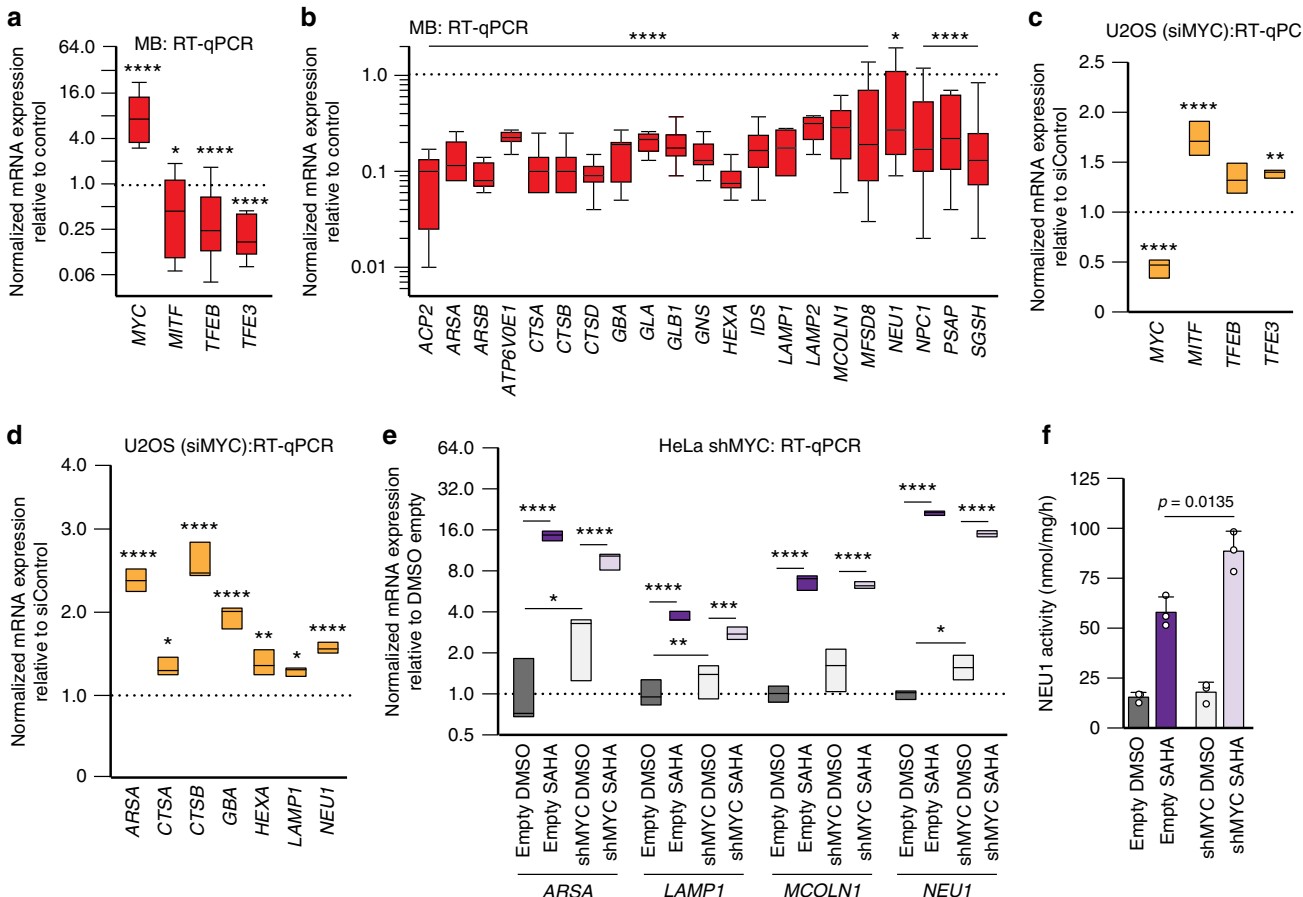

**Fig. 4** MYC represses lysosomal gene expression. **a**, **b** RT-qPCR of (**a**) *Myc*, *Mitf*, *Tfeb*, *Tfe3* and (**b**) lysosomal genes was performed in mouse group 3 medulloblastoma (MB) tumorspheres overexpressing *Myc* (*Trp53^{-/-}* overexpressing *Myc*) and values compared to *Trp53^{-/-}* controls (n = 10 biologically independent samples). **c**, **d** RT-qPCR of (**c**) *MYC*, *MITF*, *TFEB*, *TFE3* and (**d**) lysosomal genes in U2OS osteosarcoma cells with silenced MYC expression. **e** RT-qPCR of *ARSA*, *LAMP1*, *MCOLN1* and *NEU1* was performed in HeLa cells silenced for MYC (shMYC), (n = 6 biologically independent samples) and treated with SAHA (8 μM for 24 h). **f** Activity assays for NEU1 in HeLa cells silenced for MYC (shMYC) or infected with a lentiviral empty vector control (Empty) (n = 3 biologically independent samples) treated with DMSO and SAHA (8 μM for 24 h). All graphs are presented as mean ± SD. Statistical analysis was performed using the Student *t*-test. *p < 0.05, **p < 0.01, ***p < 0.001, ****p < 0.0001

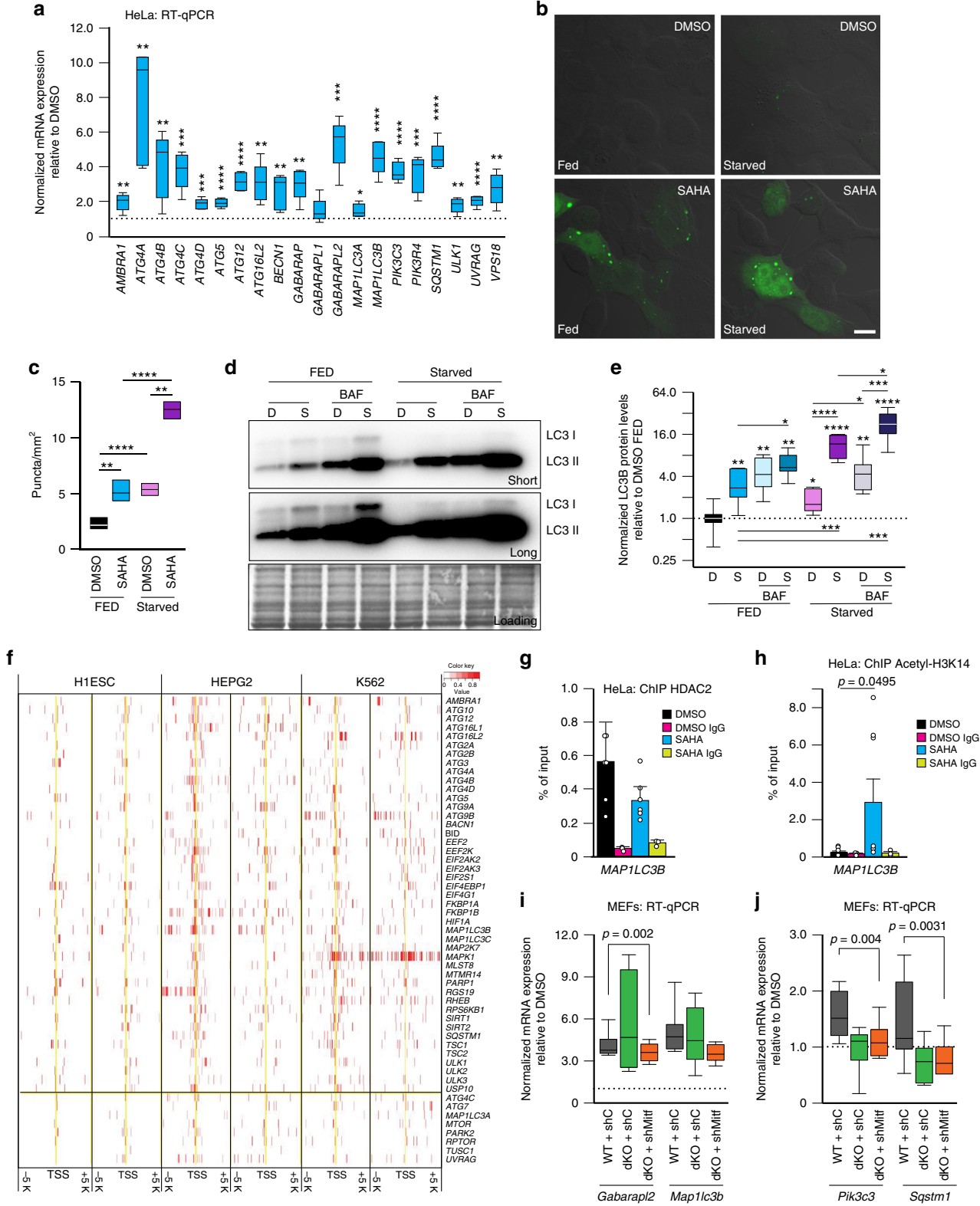

enzyme in all treated fibroblasts, which was paralleled by enhanced NEU1 residual enzyme activity (Fig. 8b, c and Supplementary Fig. 7). ChIP performed with acetylated H3K14 antibody showed acetylation of the NEU1 promoter region also in sialidosis fibroblasts, confirming NEU1 transcriptional activation (Fig. 8d). These findings potentially broaden the use of HDAC inhibitors for ameliorating the disease phenotype in some forms of LSDs.

**MYC antagonizes MiT/TFE in cancer and pluripotent stem cells.** Collectively our data suggest a rheostat model based on mutual exclusivity between MYC and MiT/TFE expression. We sought to investigate this paradigm in the context of cell proliferation and differentiation programs, which are known to be associated with high or low levels of MYC[27] We chose as model systems tumor tissue samples and human induced pluripotent stem cells (hiPSCs) reprogrammed from human

**Fig. 5** HDACs epigenetically regulate autophagy. **a** RT-qPCR of autophagy genes in SAHA-treated (20 μM for 24 h) HeLa (n = 5 biologically independent samples). **b** EGFP-LC3 expression in HeLa treated with DMSO or SAHA (20 μM for 24 h) under fed or starved conditions (EBSS). Scale bar, 20 μm. **c** Quantification of EGFP-LC3 (n = 12 images). **d** Top, representative image of anti-LC3 immunoblot of HeLa treated with DMSO (D) or SAHA (S) (20 μM for 24 h) under fed or starved conditions (EBSS) with or without Bafilomycin A1 (BAF) to assess the autophagic flux; short and long exposure. Bottom, Coomassie stained immunoblot used as loading control **e** Quantification of LC3II/LC3I ratio (n = 5 independent experiments) normalized to the loading control and relative to DMSO FED values; D = DMSO, S = SAHA, starved = EBSS treatment. **f** HDAC2 occupancy of autophagy gene promoters analyzed using ChIP-seq datasets from the ENCODE/Haib project. Genes on the left of the vertical black line are occupied by HDAC2. p = 4.656e-08, odds ratio = 6.0, Fisher's exact test. **g** ChIP analyses of the *MAP1LC3B* promoter using anti-HDAC2 antibody was performed in HeLa treated with SAHA (20 μM for 24 h) or DMSO (n = 6 biologically independent samples); IgG control (n = 3 biologically independent samples). **h** ChIP analysis of the promoter of *MAP1LC3B* using acetyl histone H3 Lys 14 antibody (Acetyl-H3K14) in HeLa treated with SAHA (20 μM for 24 h) or DMSO (n = 6 biologically independent samples); IgG control (n = 3 biologically independent samples). **i, j** RT-qPCR of (**i**) *Gabarapl2*, *Map1lc3b* and (**j**) *Pik3c3* and *Sqstm1* was performed after SAHA treatment (20 μM for 24 h) in WT MEFs, in MEFs with CRISPR-mediated knockout of *Tfeb* and *Tfe3* (dKO) and in MEFs dKO silenced for *Mitf* (dKO + shMitf) (n ≥ 6 biologically independent samples). shC refers to cells transduced with shRNA control lentivirus. Statistical analysis was performed comparing the normalized expression of the autophagy genes in MEFs WT + shC versus dKO + shC and in MEFs WT + shC versus dKO + shMitf. Boxes represent the mean value and bar inside the box represents median value; upper bar represents maximum of distribution; lower bar represents minimum of distribution (95% confidence level). Graphs are presented as mean ± SD. Statistical analysis was performed using the Student t-test. *p < 0.05, **p < 0.01, ****p < 0.0001

fibroblasts, because the processes of carcinogenesis and reprogramming share many common features: both cancer cells and hiPSCs have a sustained proliferative potential, replicative immortality, and lose their original differentiation state[50]. Furthermore, upon oncogenic activation, cancer cell progenitors acquire stem cell-like characteristics by inducing a dedifferentiation program[50]. For this purpose, we first assessed the levels of MYC and MiT/TFE proteins in a human tumor tissue microarray derived from colon adenocarcinoma, and in patient-derived xenografts from group 3 medulloblastoma and rhabdomyosarcoma. The immunohistochemistry (IHC) findings for each of the cancers showed that neoplastic cells were heterogenous in their nuclear and cytoplasmic expression of MYC, HDAC2 and TFE3. Importantly, the nuclear expression of MYC and HDAC2 were common in subpopulations of neoplastic cells, regardless of their ontogeny. On the other hand, TFE3 appeared to be more frequently localized to the cytoplasm in neoplastic cells that expressed MYC and HDAC2 in their nuclei (Fig. 9a, b and Supplementary Fig. 8a–e).

Reprogrammed hiPSCs that showed a normal karyotype and had no viral integrations (Supplementary Fig. 9a, b) were subjected to comparative proteomic profiling with their parental fibroblasts. We found that in hiPSCs the increased levels of both MYC and HDAC2 proteins were counterbalanced by a clear decrease in the levels of TFEB (Fig. 9c), a finding that explains the reduced expression of both autophagy and lysosomal proteins in hiPSCs compared to their parental fibroblasts (Fig. 9d, e). Western blot analyses performed on hiPSCs and parental fibroblasts confirmed the activation of MYC and HDAC2 (Fig. 9f and Supplementary Fig. 9c–f) and consequent decreased levels of the lysosomal NEU1 and LAMP1 (Fig. 9f and Supplementary Fig. 9g–j). Finally, correlative analysis of microarray of hiPSCs at different reprogramming stages[51] demonstrated that the expression levels of *FOXH1* positively correlated with those of a subset of autophagy genes (Supplementary Table 4). Together, these data suggest that HDAC/MYC and MiT/TFE or FOXH1 define distinct cellular fates and are expressed in a mutually exclusive manner to control lysosomal biogenesis and autophagy.

## Discussion

The notion that activation or repression of lysosomal biogenesis and autophagy could be simultaneously controlled by epigenetic and transcriptional mechanisms has been hypothesized, but until now, no evidence has been shown for such coordinated regulation. In this study, we uncovered a hierarchical and dynamic rheostat that couples HDACs with MYC to

epigenetically and transcriptionally regulate these cellular catabolic machineries. By modulating the levels of available MYC, HDACs bestow an inhibitory loop on lysosomal and autophagy genes. Whether this is because HDACs regulate MYC transcription, as our current data seem to suggest, or because knockdown of HDACs leads to decreased cell proliferation and therefore MYC is downregulated, remains a question to be addressed in future studies. In either case, the effect of reduction of MYC levels is to free the targeted promoter sites of lysosomal and autophagy genes, thereby allowing the binding by MiT/TFE members or FOXH1, and activation of their transcription. This model may explain how lysosomal/ autophagy function changes within a cell, and how cells transition and adapt through different metabolic states. Furthermore, our data suggest that this epigenetic rheostat has important implications for the treatment of LSDs. Thus, changes in the concentrations of HDACs, MYC, and MiT/TFE in LSDs may contribute to the clinical and phenotypic heterogeneity of these disorders. In agreement with this hypothesis, 11 HDAC genes have been found upregulated in fibroblasts from patients with Niemann-Pick Type C disease[12].

Until the discovery of MiT/TFE involvement in lysosomal biogenesis, lysosomes were mostly considered the recycling/ degradation center of the cell, and their housekeeping-type genes were thought to require little or no regulation. MiT/TFE studies demonstrated the existence of a broader network of transcription factors, regulating mainly autophagy function. In addition, a large body of literature has focused on how modifications of histones regulate autophagy[52,53]. However, none of these studies addressed the simultaneous regulation of these two catabolic machineries that we have now investigated. Our results demonstrate the concurrent involvement in these processes of transcription factors and nucleus-residing HDACs. Several HDACs are known to regulate autophagy, but not lysosomal biogenesis. However, their effects are cell-type specific and mainly involve cytosolic acetylation of targeted proteins, rather than acetylation of histones in the nucleus[54,55]. In support of a nuclear control of autophagy, inhibition of HDAC1 has been shown to induce autophagy and to promote cell death in several types of cancer cells[56,57]. Our proposed mode of autophagy regulation by antagonistic transcription factors may share similarities to the "autophagy master switch"[58], and the "hepatic nutrient-sensing" models[59] previously proposed. In our model, however, MYC not only opposes the activity of MiT/TFE on lysosomal and autophagy gene promoters, but also modulates their levels in an acetylation-dependent manner.

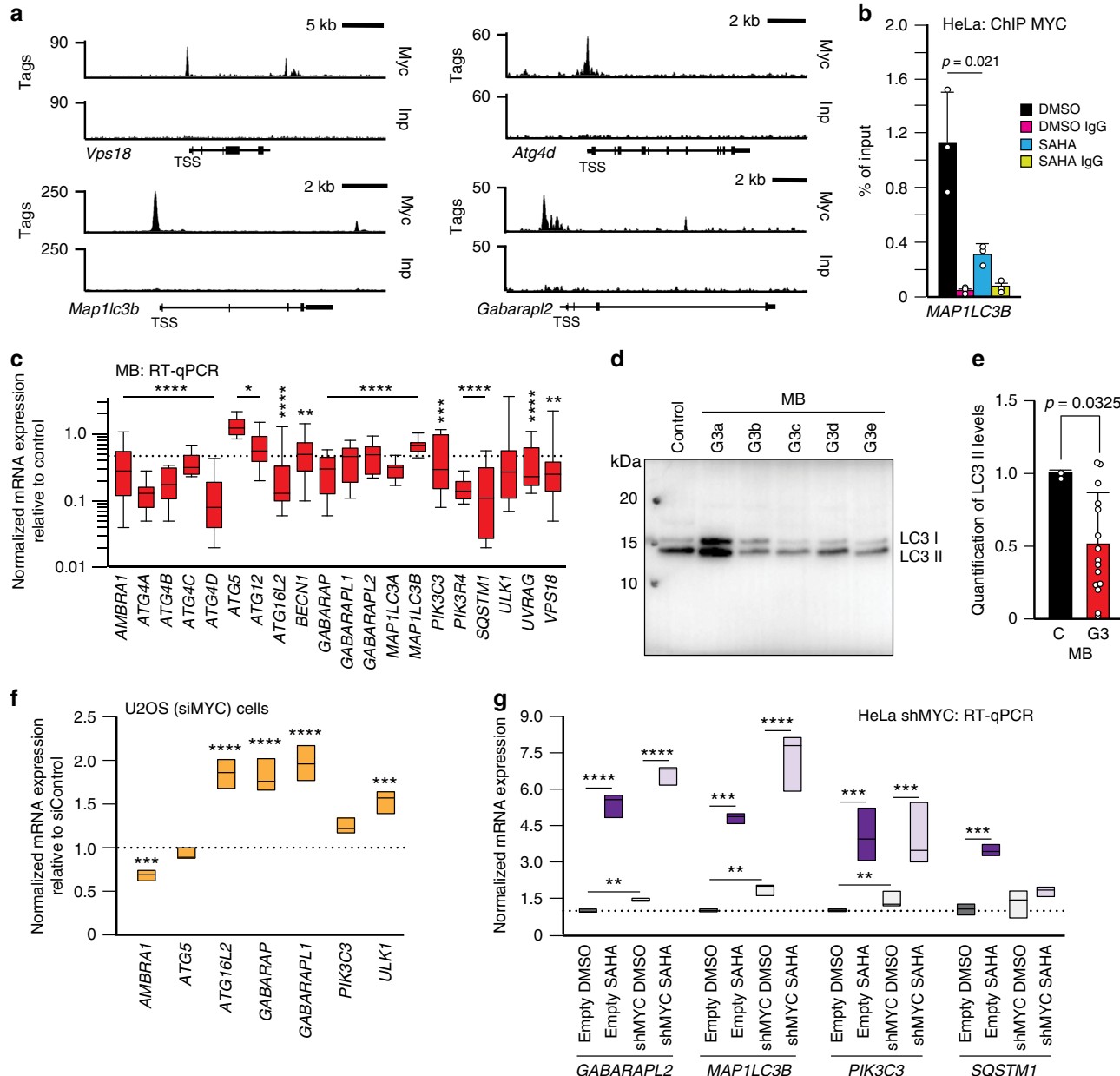

**Fig. 6** MYC represses autophagy. **a** ChIP-seq analysis for Myc binding to the promoters of *Vps18, Map1lc3b, Atg4d*, and *Gaparapl2* performed in mouse group 3 medulloblastoma tumorspheres overexpressing *Myc*, MB (*Trp53*−/− overexpressing *Myc*) (*n* = 3 independent experiments). **b** ChIP analyses of the *MAP1LC3B* promoter using anti-MYC antibody performed in HeLa cells treated with SAHA (20 μM for 24 h) or DMSO (*n* = 3 biologically independent samples). **c** RT-qPCR of autophagy genes in tumorspheres Trp53−/− overexpressing Myc (MB), (*n* = 10 biologically independent samples). **d** Comparative LC3 immunoblots of tumorspheres *Trp53*−/− overexpressing *Myc* and control *Trp53*−/− cells. **e** Quantification of the LC3II/LC3I ratio normalized to Coomassie stained immunoblots, used as loading control. C = Control (*Trp53*−/−) cells (*n* = 3 biologically independent samples); G3 = tumorspheres *Trp53*−/− overexpressing *Myc* (*n* = 15 biologically independent samples). **f** Expression levels of several autophagic genes in U2OS cells with silenced MYC expression. **g** RT-qPCR of *GABARAPL2, MAP1LC3B, PIK3C3* and *SQSTM1* was performed in HeLa cells silenced for MYC (shMYC) (*n* = 3 biologically independent samples) and treated with SAHA (8 μM for 24 h). Boxes represent the mean value and bar inside the box represents median value; upper bar represents maximum of distribution; lower bar represents minimum of distribution (95% confidence level). Graphs are presented as mean ± SD. Statistical analysis was performed using the Student *t*-test. *\*p < 0.05, \*\*p < 0.01, \*\*\*p < 0.001, \*\*\*\*p < 0.0001*

The discovery that FOXH1 regulates autophagy increases the number of transcription factors that control this process, although their interplay and response to different intrinsic or extrinsic cues remains to be determined. FOXH1 belongs to the FOXO family of transcription factors, which are pivotal during development and are emerging as key regulators of autophagy. One member, FOXO3, was the first transcription factor found to control autophagy[60]. Like the MiT/TFE family members, FOXOs are regulated by phosphorylation and shuttle between the cytoplasm and the nucleus, where they activate or inhibit autophagy genes[60,61]. FOXO1, one of the most abundant FOXOs, can induce autophagy in the cytosol by directly binding to autophagy-related proteins[62]. This transcription factor has also been shown to prevent cellular dedifferentiation during early endocrine development[63]. Considering the role of FOXH1 in patterning the anterior primitive streak and in nodal signaling[47,64], the possible

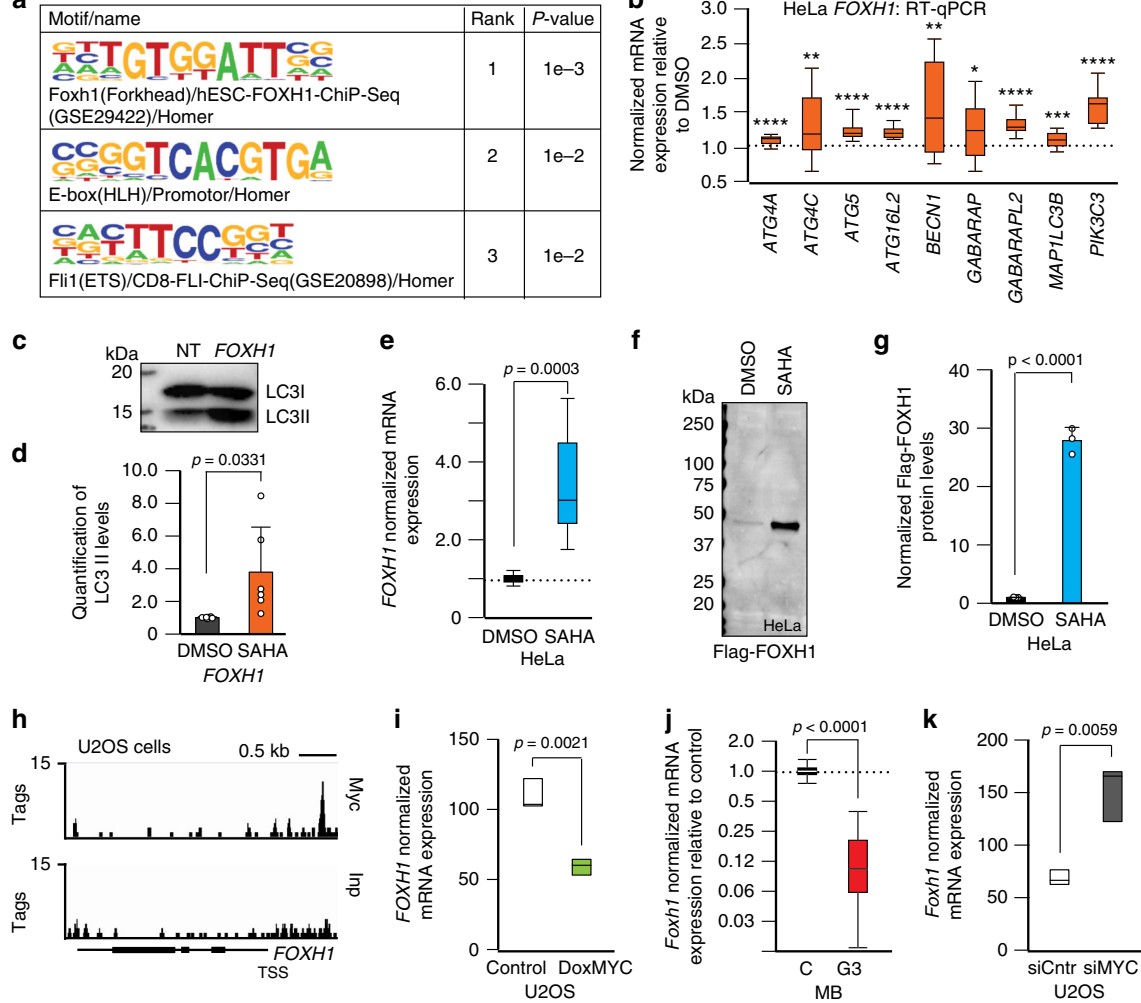

**Fig. 7** FOXH1 is an additional activator of autophagy and responds to the HDACs/MYC axis. **a** Motif analysis using HDAC2-binding sites for autophagy genes identified the FOXH1 motif. **b** RT-qPCR of autophagy genes in FOXH1-overexpressing HeLa cells ($n = 6$ biologically independent samples) $p < 0.01$. **c** Representative LC3 immunoblot from FOXH1-overexpressing cells. **d** Quantification of the LC3II/LC3I ratio normalized to Coomassie stained immunoblots, used as loading control ($n = 6$ independent experiments). **e** *FOXH1* RT-qPCR in SAHA-treated HeLa cells (20 μM for 24 h; $n = 7$ biologically independent samples). **f** Representative immunoblot of FOXH1 overexpression in HeLa cells treated with DMSO or SAHA (20 μM for 24 h). **g** Quantification of **f** ($n = 3$ independent experiments) normalized to Coomassie stained immunoblots, used as loading control. **h** MYC binding to the *FOXH1* locus obtained from ChIP-seq datasets for U2OS cells overexpressing MYC. **i** Expression of the *FOXH1* gene in U2OS cells with MYC overexpression, data obtained from Walz et al. **j** RT-qPCR of *Foxh1* in tumorspheres *Trp53*$^{-/-}$ overexpressing *Myc*. Values are relative to those obtained in control *Trp53*$^{-/-}$ cells ($n = 5$ biologically independent samples). **k** RT-qPCR of *FOXH1* in U2OS cells in which MYC was silenced by shRNA. Boxes represent the mean value and bar inside the box represents median value; upper bar represents maximum of distribution; lower bar represents minimum of distribution (95% confidence level). Graphs are presented as mean ± SD. Statistical analysis was performed using the Student *t*-test

involvement of this transcription factor in embryonic specification via regulation of autophagy merits future investigation.

We also uncovered a previously uncharacterized antagonistic role between MYC and MiT/TFE that controls gene expression programs during cell differentiation. The proposed mechanism implies that the acquisition of "stemness" requires robust activation of MYC and the active repression of lysosomal and autophagy genes. In contrast, the maintenance of a fully differentiated state requires activation of lysosomal and autophagy signatures by MiT/TFE members. In agreement with this hypothesis, we observed upregulation of MYC and downregulation of TFEB in hiPSCs when compared to fully differentiated isogenic fibroblasts. The mutually exclusive expression of these factors, likely regulated by HDAC2, ensures repression of the lysosomal and autophagic systems in hiPSCs. Although it has been amply demonstrated that induction of pluripotency

correlates with the presence of acetylated, transcriptionally-permissive chromatin, our results suggest a stage-specific requirement for HDAC2 in reprogramming, in line with a recent publication demonstrating that HDAC2 inhibits the early phase of reprogramming, but positively affects the final stages of this process[65].

Cellular reprogramming shows remarkable parallels with the formation of cancer stem cells[50]. The factors necessary for somatic reprogramming are also oncogenic drivers and promote the re-acquisition of early developmental stages and unlimited proliferation[66]. MYC, for instance, is required to potently generate fully reprogrammed hiPSCs[67–69]. It is also overexpressed in more than 70% of human tumors, driving tumorigenesis and maintenance[70]. Blocking MYC activity causes tumor regression by promoting cell cycle arrest and differentiation[71]. Our studies support the notion that tumors may use the HDAC/MYC–MiT/

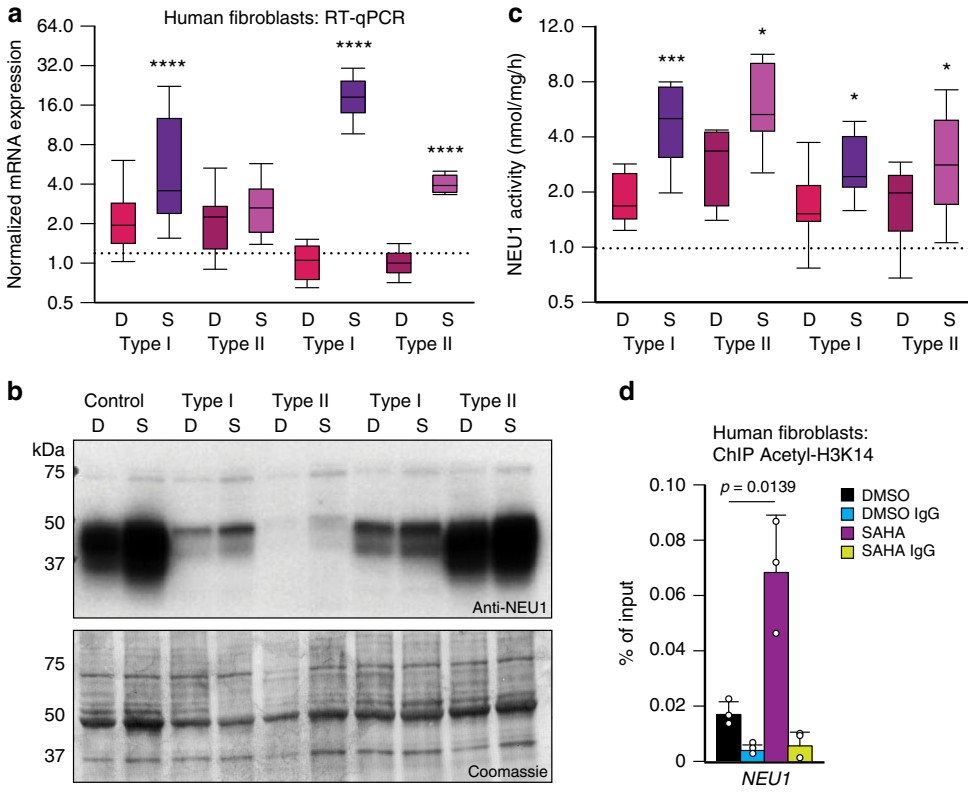

**Fig. 8** HDAC inhibition in LSD. **a** RT-qPCR of *NEU1* in SAHA- or DMSO-treated sialidosis fibroblasts, (SAHA 20 μM for 24 h; *n* = 8 biologically independent samples). Type I = attenuated form of sialidosis; Type II = severe form of sialidosis. **b** Top, representative immunoblot of fibroblasts isolated from two patients with sialidosis type I and two patients with sialidosis type II were treated with SAHA (S) (20 μM for 24 h) or DMSO (D) and probed with anti-NEU1 antibody; bottom, Coomassie stained immunoblot used as loading control. **c** NEU1 activity in fibroblasts isolated from two patients with sialidosis type I and two patients with sialidosis type II treated with SAHA (S)- or DMSO (D)-treated (SAHA, 20 μM for 24 h; *n* = 7 biologically independent samples). **d** ChIP analysis of the promoter of *NEU1* performed with acetyl histone H3 Lys 14 antibody (Acetyl-H3K14) in fibroblasts from one type I sialidosis fibroblasts treated with SAHA (20 μM for 24 h) or DMSO (*n* = 3 biologically independent samples). Boxes represent the mean value and bar inside the box represents median value; upper bar represents maximum of distribution; lower bar represents minimum of distribution (95% confidence level). Graphs are presented as mean ± SD. Statistical analysis was performed using the Student *t*-test. *$p < 0.05$, ***$p < 0.001$, ****$p < 0.0001$

TFE axis to repress their lysosomal and autophagy systems, a paradigm that may be exploited therapeutically. In fact, HDAC inhibitors have recently shown therapeutic potential in group 3 medulloblastoma, as a single-agent therapy[72] or in combination with PI3K inhibitors[73]. Given that HDAC inhibitors increase FOXO1 expression, thereby inhibiting tumor growth, and that silencing FOXO1 dampens this effect, we are tempted to speculate that autophagy and lysosomal biogenesis activated by MiT/TFE and/or FOXH1 may be the pathways of choice for targeted cancer therapy. Future studies will reveal the generality of our rheostat model in different cancer types, and how it contributes to tumor initiation, progression and maintenance.

## Methods

**Cell culture and treatment.** Human cell lines HeLa and RH30 are available at the ATCC and were provided by the laboratory of Dr. Grosveld. Cells were grown at 37 °C in Dulbecco's modified Eagle's medium (DMEM) supplemented with 10% cosmic calf serum (HyClone), 2 mM GlutaMAX, penicillin (100 U/mL), and streptomycin (100 μg/mL) (Gibco) in a humidified 5% $CO_2$ atmosphere. Neuroblastoma cells (Sy5y), were grown in a 1:1 mixture of F12 (Gibco) and Eagle's minimal essential medium (EMEM) (Lonza), both supplemented with 10% fetal bovine serum, GlutaMAX, and antibiotics, as indicated for HeLa cells. Tumor-spheres from mouse group 3 medulloblastoma (*Trp53*⁻/⁻ overexpressing *Myc*) (#19251; #19554, #15486; #13465; #19568)[40,44], and tumorspheres from *Trp53*⁻/⁻ (used as control) were grown on ultralow-attachment plates and supplemented with human recombinant basic FGF and EGF (Peprotech) every 3 days. MEFs WT and MEFs with CRISPR-mediated ablation of Tfeb and Tfe3 were grown in Dulbecco's modified Eagle's medium (DMEM) supplemented with 10% primary

serum, 2 mM GlutaMAX, penicillin (100 U/mL), and streptomycin (100 μg/mL) (Gibco) in a humidified 5% $CO_2$ atmosphere with 2 μg/mL Puromycin.

Skin fibroblasts from control individuals were obtained from Coriell Institutes; human sialidosis fibroblasts type I or type II received by my laboratory were uncoded and unidentifiable. They came from the Rotterdam Biobank (Rotterdam, The Netherlands), the Pediatric Undiagnosed Diseases Program, National Human Genome Research Institute/NIH (Bethesda MD, USA) and the Muscle Unit Section and Laboratory of Skeletal Muscle Pathology Department of Neurology, Medical School of the University of São Paulo (São Paulo, Brazil). Original consent was obtained by the clinicians from the patient or a family member and the study was approved by the ethics committees of the three institutions. All cells were banked for secondary future research.

For transient expression, HeLa cells were plated on 4-chambered Lab-Tek glass coverslips (Nalge Nunc Int.) and transfected with the FOXH1 plasmid[74] and with the EGFP-LC3 plasmid using the calcium phosphate precipitation method. Transfected cells were washed with PBS 3 times and fixed for 10 min in PBS containing 4% PFA and then washed extensively after fixation. Slides were mounted with ProLong Antifade Mountants (Thermofisher) and imaged using a confocal fluorescence microscope (Zeiss LSM 780). LC3 positive autophagosome were counted with the Zeiss software in more than 10 fields; ROI(s) or "objects" were determined for each image. The number of objects and the area of the ROI were recorded and calculated as puncta/mm2. Autophagic flux was assayed probing western blots with anti-LC3 antibody in HeLa cells treated with bafilomycin A1 (Sigma) under FED (complete medium) or Starved conditions. In the latter case, HeLa cells were washed extensively with a Hanks' balanced salt solution (HBSS) (Invitrogen) and incubated for 3–6 h at 37 °C in Earle's balanced salt solution (EBSS) (Sigma-Aldrich). HeLa cells were transfected with two short hairpin RNA (shRNA) constructs for human *HDAC2 and MYC* genes cloned into the pLKO1 HIV-based lentiviral vector (Dharmacon). MEFs with CRISPR-mediated ablation of Tfeb and Tfe3 were transduced with the GIPZ microRNA-adapted shRNA for Mitf (Dharmacon).

For lentivirus preparation, 293T cells were co-transfected of with 10 μg of pCAGkGP1R and 2 μg of pCAG4RTR2 packaging plasmids, 2 μg of pCAG-VSVG

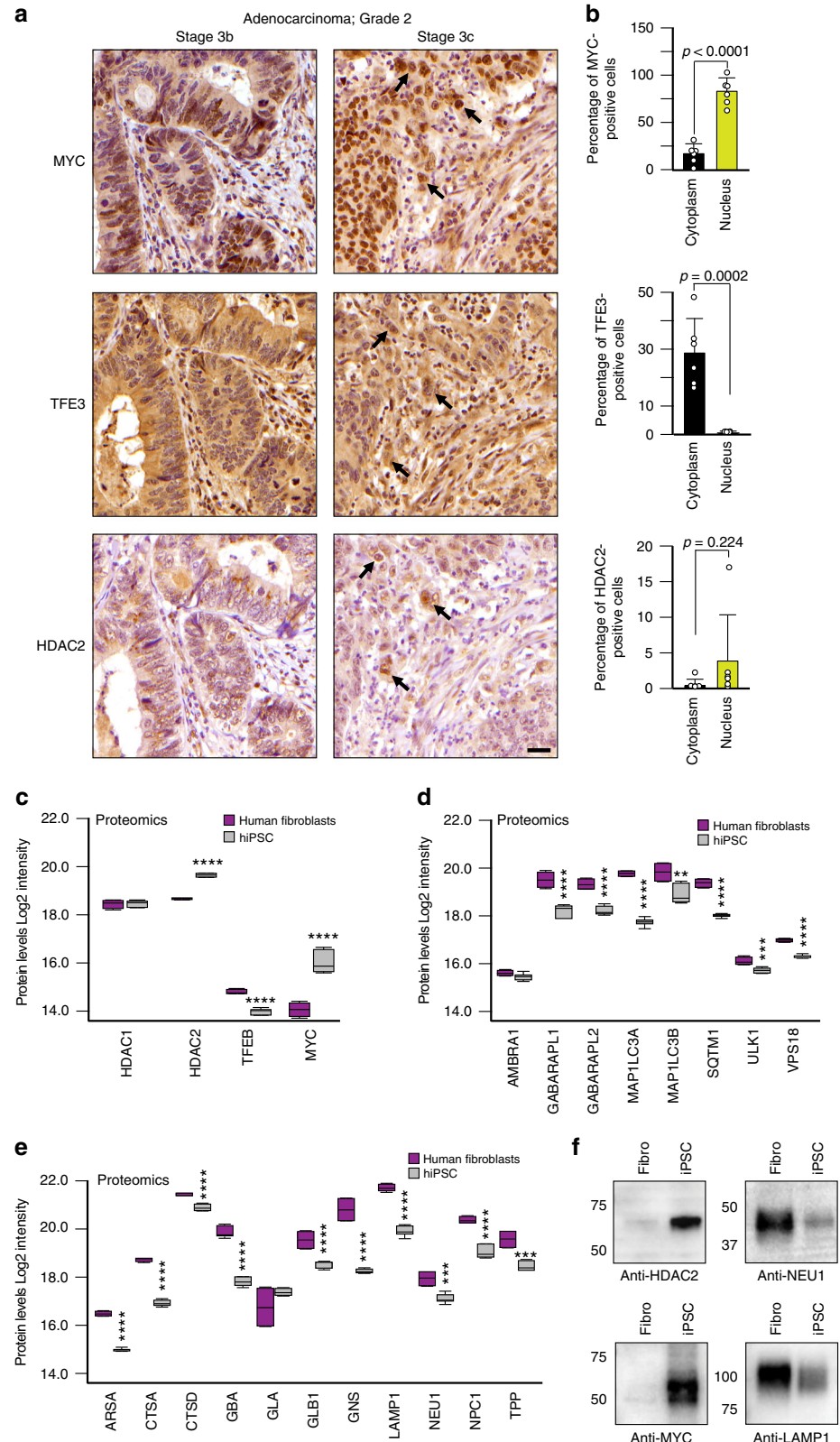

envelope plasmid, and 10 μg of vector plasmid (shHDAC2, shMYC), or 7 μg of psPAX2 (a gift from Didier Trono, Addgene plasmid # 12260; http://n2t.net/addgene:12260; RRID:Addgene_12260), 2 μg pCAG-VSVG envelope plasmid and 10 μg of vector plasmid (shMitf, Dharmacon). An empty vector or an shRNA control vector were used as controls. Stable cells were selected with puromycin (2 μg/mL, Sigma- Aldrich) or in the case of GIPZ with FACS. Cells were analyzed 72 h after transfection. For SAHA treatment, cells were incubated for 24 h at 37 °C in medium containing 20 μM or when specified 8 μM SAHA (Sigma-Aldrich)

diluted in DMSO (Sigma-Aldrich). For romidepsin treatment, cells were incubated for 24 h at 37 °C in medium containing 10 nM romidepsin (Active Motif) diluted in DMSO (Sigma-Aldrich). Medium containing DMSO, without SAHA or romidepsin, was used as control.

**Western blot analysis and immunoprecipitation**. Cells were lysed in RIPA buffer [0.1% sodium dodecyl sulfate (SDS), 1% sodium deoxycholate, 1% Triton X-100,

**Fig. 9** HDACs/MYC–MiT/TFE axis in cancer and pluripotent stem cells. **a** Immunohistochemical analysis of MYC, TFE3, and HDAC2 in colon carcinoma samples. Regions marked by thick black arrows show that the nuclear expression of MYC and HDAC2 are common in subpopulations of neoplastic adenocarcinoma cells, especially at the invasive edges of the tumors. TFE3 appears to be more frequently localized to the cytoplasm in these same cells. Scale bar 25 μm. **b** MYC, TFE3, and HDAC2 expression levels in the cytoplasm and the nucleus of neoplastic cells were scored separately ($n = 6$ independent tumors). **c**–**e** Protein levels of (**c**) HDAC1, HDAC2, TFEB, and MYC, (**d**) of lysosomal enzymes and lysosomal membrane components and (**e**) of autophagy effectors detected in proteomics of human fibroblasts ($n = 4$ biologically independent samples) and reprogrammed hiPSCs, ($n = 7$ biologically independent samples). **f** Representative immunoblots of hiPSCs and their parental fibroblasts probed with anti-MYC, anti-HDAC2, anti-NEU1 and anti-LAMP1 antibodies. Boxes represent the mean value and bar inside the box represents median value; upper bar represents maximum of distribution; lower bar represents minimum of distribution (95% confidence level). Graphs are shown as mean ± SD. Statistical analysis was performed using the Student $t$-test. $^{**}p < 0.01$, $^{***}p < 0.001$, $^{****}p < 0.0001$

140 mM NaCl, 10 mM Tris, protease inhibitors cocktail, and phosphatase inhibitors]. Protein concentration was determined by optical density ($A_{595}$) using the Pierce bovine serum albumin (BSA) protein assay (Pierce). For immunoprecipitation, HeLa cells were lysed in RIPA buffer with protease inhibitors cocktail, and phosphatase inhibitors (Roche). Total protein lysates (250 µg) were incubated overnight at 4 °C on a rotating platform with 2 µg of anti-MYC antibody (Cell Siganling) or normal immunoglobulin G (IgG) (Cell Signaling). Dynabeads Protein G (Millipore) were simultaneously incubated with V5 tag (1 µg/ml) antibody–blocking peptide. Beads were washed extensively with RIPA buffer and added to the cell lysates; samples were incubated for an additional hour. The beads were then washed four times with RIPA buffer. Proteins were eluted with sample loading buffer before immunoblotting. Immunoprecipitates were run on 4-20% Bis-Tris mini protein gels (Biorad) and transferred ON. Membranes were immunoblotted for anti-HDAC2 antibody (Cell Signaling #2545, 2µg) and anti-MYC antibody (Cell Signaling 9402, 2µg). For immunoblot analyses, HeLa, RH30, Sy5y and medulloblastoma cells were lysed in RIPA buffer; MEFs with CRISPR-mediated ablation of Tfeb and Tfe3 were lysed in 25 mM Hepes-KOH, pH 7.4, 150 mM NaCl, 5 mM EDTA, and 1% Triton X-100 (w/v) supplemented with protease and phosphatase inhibitors (Roche). MEF lysates were incubated on ice for 30 min, mechanically sheared through a 25-gauge needle. and centrifuged at $16,000 \times g$ for 10 min at 4 °C. Protein lysates (20-35 µg) underwent electrophoresis on NuPage Novex (4–12%), 10, 12, and 4–20% Bis-Tris mini protein gels (Life Technologies or Biorad) and were wet-blotted onto PVDF membranes. Membranes were blocked in Tris-buffered saline containing 0.1% Tween-20 (TBS-T) and 5% nonfat dry milk and incubated overnight at 4 °C with primary antibodies in either 3% BSA in TBS-T solution or 5% milk in TBS-T. The following day, membranes were incubated with HRP-conjugated or fluorescent-tagged secondary antibodies and developed using SuperSignal West Femto Maximum Sensitivity Substrate (Thermo Scientific). The following antibodies were used for all the immunoblot analyses: anti-MYC (Cell Signaling 9402, 1:1000), anti-Lamp1 (BD 553792, 1:500), anti-LAMP1 (Cell Signaling 9091, 1:1000), anti-LC3B (Cell Signaling 3868, 1:1000), anti-Flag M2 (Sigma F1804 1:500), anti-HDAC2 (Abcam 7029, 1:1000, Cell Signaling 2545, 1: 1000), anti-TFEB (Bethyl laboratories Inc A303-672A, 1:2000) and anti-TFE3 (Sigma-Aldrich HPA023881, 1:1000) Anti-NEU1 and anti-PPCA antibodies were generated in-house.

Uncropped and unprocessed scans of the immunoblots shown in Fig. 2d, e are included in Supplementary Fig. 4c, d; those shown in Fig. 5d are included in Supplementary Fig. 5l; those shown in Fig. 7c are included in Supplementary Fig. 6c.

**Enzymatic activities**. NEU1, β-hexosaminidase, α-mannosidase and β-galactosidase activities were assayed using the synthetic substrates, 2′-(4-methylumbelliferyl)-α-D-N-acetylneuraminic acid; sodium salt, 4MU-N-acetyl- β-D-glucosaminide, 4MU-α-D-mannopyranoside, and 4 Methylumbelliferryl β-d-Galactopyranoside, respectively. Briefly, cells were collected and lysed in water. Lysates were incubated with the substrate in triplicate in 96-well plates for 1 h at 37 °C. To stop the enzymatic reaction, 200 µL 0.5 M carbonate buffer, pH 10.7, was added to all wells. The fluorescence was measured on a plate reader (EX-355, EM-460). The net fluorescence values were compared with those of the linear 4MU standard curve and were used to calculate the specific enzyme activities. Activities were calculated as nanomoles of substrate converted per hour per milligram of protein (nmol/mg/h).

For cathepsin A activity, the synthetic substrate N-carbobenzoxy-L-phenylalanyl-L-alanine was used. Briefly, cells were collected and lysed in water and cell homogenates (10 µL) were incubated with 100 µl MES Buffer for 1 h at 37 °C in the absence or presence of 1.5 mM N-blocked dipeptide Z-Phe-Ala, (10 µL). The reaction was stopped by heating the plate at 100 °C for 5–10 min following cooling on ice for 5 min. 10 µL from each well was transferred to a new 96-well plate and 250 µL Buffered Reagent (o-phtaldialdehyde/2-mercaptoethanol/ethanol solution in 50 mM Na-Carbonate buffer pH9.5) was added to each well and incubated for 5 min before measuring the fluorescence (Ex-355, Em-460). Values were calculated as subtraction between values obtained in presence of substrate minus values attained without substrate and expressed as nmol/mg/min.

**Microarray analysis**. Total RNA (100 ng) extracted from DMSO- and SAHA-treated HeLa cells (three biological replicates of each condition) was converted into biotin-labeled cRNA (Ambion WT Expression Kit, Affymetrix Inc) and hybridized to a Human Gene 2.0 ST GeneChip (Affymetrix Inc). Probe signals from arrays were normalized and transformed into $\log_2$ transcript expression values using the Robust Multiarray Average algorithm (Partek Genomics Suite v6.6). GSEA (http://software.broadinstitute.org/gsea/index.jsp) was performed using curated pathways obtained from MolSigDB (Broad Institute). Differentially expressed transcripts were identified by ANOVA and the false discovery rate (FDR) was estimated. Functional enrichment analysis of gene lists was performed using the DAVID bioinformatics databases (http://david.abcc.ncifcrf.gov/).

**Immunohistochemistry**. Immunohistochemistry (IHC) analyses were performed on 10-µm-thick paraffin sections obtained from one RH30R xenograft, one medulloblastoma subgroup 3 xenograft and from a tissue microarray containing 6 cases of colon adenocarcinoma (TMA CO243b, US Biomax, Inc.). After the blocking solution (0.1% BSA, 0.5% Tween-20, and 10% normal serum) step, sections were incubated overnight with the specific antibody diluted in blocking buffer (anti-MYC, Cell Signaling 9402, 1:300, anti-TFE3 Sigma HPA023881 1:300 and anti-HDAC2, Abcam ab7029 1:500). The sections were washed and incubated with biotinylated secondary antibody (Jackson ImmunoResearch Laboratory) for 1 h. Endogenous peroxidase was removed by incubating the sections with 0.1% hydrogen peroxidase in PBS for 30 min. Antibody detection was performed using the ABC Kit and diaminobenzidine substrate (Vector Laboratories) and sections were counterstained with haematoxylin according to standard method. The stained slides were scanned with an Aperio ScanScope XT scanner (Leica Biosystems) to create whole slide images that were scalable up to a magnification of ×200. Protein marker expression was scored separately for the cytoplasm and the nucleus (Aperio ImageScope software, Leica Biosystems). A manual review of all IHC slides was also performed using a Nikon Eclipse Ni microscope and correlated with the algorithm generated data. Findings were reported as the percentage of subpopulations of neoplastic cells per subcellular location based on a combination of the automated and manual assessments.

**Chromatin immunoprecipitation**. Each chromatin immunoprecipitation (ChIP) analysis was performed with $5 \times 10^6$ cells by following a published procedure[75]. Cells were cross-linked with 1% formaldehyde for 10 min at room temperature, and the reaction was stopped by adding glycine (final 125 mM) for 5 min. The cross-linked cells were lysed in 800 µL buffer (50 mM Tris pH 8.1, 10 mM EDTA, 1% SDS), supplemented with a protease inhibitor cocktail (Roche). After dilution in 2.0 mL IP buffer (25 mM Tris pH 8.0, 150 mM NaCl, 0.01% SDS, 0.5% deoxycholate, 1% TritonX-100, and 5 mM EDTA), cell lysates were sonicated to generate DNA fragments of 200- to 600-bp. Fragmented chromatin was immunoprecipitated with salmon sperm DNA/protein A agarose beads (Millipore) with specific antibodies. The antibodies used for the ChIP experiments were: anti-HDAC2 (Abcam ab7029, 5µg), anti-MYC (Cell Signaling 9402, 1:50), anti-TFEB (Cell Signaling 4240, 1:200), anti-TFE3 (Sigma HPA023881, 5µg), anti-Acetylated Histone 3 K14 (Cell Signaling 7627 1:50). Agarose beads were also incubated with normal IgG (Cell Signaling 3900). The pulled-down immune-complexes were washed twice each with low-salt buffer (50 mM Tris pH 8.0, 150 mM NaCl, 0.1% SDS, 0.5% deoxycholate, 1% NP40, 1 mM EDTA), high-salt buffer (50 mM Tris pH 8.0, 500 mM NaCl, 0.1% SDS, 0.5% deoxycholate, 1% NP40, 1 mM EDTA), LiCl wash buffer (50 mM Tris pH 8.0, 250 mM LiCl, 0.5% deoxycholate, 1% NP40, 1 mM EDTA), and Tris-EDTA buffer (10 mM Tris pH 8.0, 1 mM EDTA) for 10 min at 4 °C. Immune-complexes were eluted in elution buffer (1% SDS, 0.1 M NaHCO₃) 2 times for 15 min and reverse cross-linked at 65 °C overnight with 5 M NaCl (Sigma-Aldrich). The nest day, the eluates were treated with 0.5 M EDTA (Millipore, 20-158), 1 M Tris-HCl, pH 6.5 (Millipore, 20-160) and 10 mg/mL Proteinase K (Sigma-Aldrich) and incubated for one hour at 45 °C. Reverse cross-linked DNA was purified by QIAquick PCR purification kit (Qiagen). ChIPs were performed at least 3 times for each of the antibodies. For sequential ChIP assays, chromatin preparations were first incubated with anti-MYC antibody (Cell Signaling 9402, 1:50); then chromatin immune-complexes were eluted with 10 mM DTT at 37 °C for 30 min, followed by the second ChIP with antibody for HDAC2

(Abcam ab7029, 5 µg). Eluted DNA was then purified and analyzed with standard RT-qPCR procedure. RT-qPCR of immunoprecipitated chromatin was carried out on a CFX96 real-time PCR Machine (BioRad) using RT2 Sybr Green qPCR Mastermix, 2 µL of the immunoprecipitated material or the purified input were used in the reactions with oligonucleotides available from SA Bioscience (Qiagen). Oligos designed to cover a region at +5 Kb from the promoter of NEU1 (NEU1 INT) were used as negative control to amplify a region for no antibody binding. The values shown in the graphs are expressed as a percentage of the total input DNA.

**Quantitative real-time PCR**. Total RNA was isolated from DMSO- and SAHA-treated HeLa, RH30, and Sy5y cell lines, romidepsin-treated HeLa and primary fibroblasts from healthy individuals, sialidosis fibroblasts, as well as from Flag-FOXH1–transfected HeLa cells by using the PureLink RNA kit (Life Technologies) following the manufacturer's protocol. DNA contaminants were removed on a DNAse I column (Life Technologies), according to the manufacturer's protocol. RNA quantity and purity were measured using a Nanodrop Lite spectrophotometer (Thermo Scientific). Complementary DNA was produced using 2 µg total RNA with RT2 First Strand Kit (Qiagen; 330401). RT-qPCR was performed using the RT² Sybr Green qPCR Mastermix, 1 µL (50 ng) cDNA, 10 µM primer, and RNAse-free water in a 25-µL reaction volume on a CFX96 real-time PCR machine (BioRad). The oligonucleotides used for RT-qPCR were purchased from Biorad. Samples were normalized by using HRTP1 (human) or 18S ribosomal RNA (mouse) expression. The plotted values represent the relative normalized expression of the mRNA in SAHA-treated and in romidepsin-treated cells. DMSO-treated cells were used as control for normalization of the expression for SAHA- and romidepsin-treated cells. RT-qPCR values obtained in tumorspheres from mouse group 3 medulloblastoma ($Trp53^{-/-}$ overexpressing $Myc$) (#19251; #19554, #15486; #13465; #19568) are shown as relative to gene expression values obtained from tumorspheres from $Trp53^{-/-}$.

**Lysosomal volume**. The lysosomal volume was analyzed as previously described[76]. Lysotracker green DND 26 (200 nM, Molecular Probes) was added to cells for 40 min. After incubation, cells were washed with PBS, counted, and analyzed by fluorescence-activated cell sorting (FACS). Mean intensities of lysotracker green fluorescence were calculated and plotted.

**Transmission electron microscopy and counting**. Transmission electron microscopy analyses were performed at the Electron Microscopy Division of the Cell and Tissue Imaging Center at St. Jude Children's Research Hospital. HeLa cells treated with DMSO or SAHA were fixed in 2.5% glutaraldehyde in 0.1 M sodium cacodylate buffer. The samples were post-fixed in 2% osmium tetroxide and dehydrated via a graded series of alcohol, cleared in propylene oxide, embedded in epon araldite and polymerized overnight at 70 °C. The unstained sections were imaged on a JEOL 1200 EX Transmission Electron Microscope with an AMT T2K digital camera. More than 20 images were taken from two biological replicates and counted blindly for the number of lysosomes.

**Reprogramming of fibroblasts into hiPSCs**. Human fibroblast (BJ and PCS201 which were male and female wild type, respectively) cells were purchased from ATCC and were grown in DMEM + 10% FBS media. Once cells were confluent, reprogramming was performed by introducing sendai viral vectors (CytoTune 2.0, Invitrogen) expressing four human reprogramming factors (Oct3/4, Sox2, Klf4, and c-Myc). In brief, ~5 × 10^5 fibroblast cells were transduced by Sendai virus. Each ESC-like colony was picked-up manually by glass pipette under the microscope and maintained on feeder-free Geltrex coated plates to establish hiPSC lines. After at least five passages of expansion, candidate clones for hiPSC lines were characterized by expression of pluripotency marker (Oct4, Santa Cruz Biotechnologies, sc-5279, clone No. C-10, 1:500; Nanog, Santa Cruz Biotechnologies, sc-33759, 1:500; SSEA4, Millipore, MAB4304, 1:500; SSEA3, Millipore, MAB4303-I, 1:500; TRA-1-60 Millipore, MAB4360, 1:500) using an immunofluorescence assay, followed by karyotype analysis. For the immunofluorescence assay, each hiPSC line was fixed immediately with fixing solution (4% paraformaldehyde) and then treated with permeabilization buffer (0.2% Triton X-100) after washing with PBS. Cells were washed with PBS three times and incubated with blocking solution which contained 3% BSA in PBS for 15 min. Cells were incubated with anti-Oct4, anti-SSEA4 and anti-SSE3A antibodies in blocking solution overnight. Cells were washed with PBS three times and incubated with secondary antibodies (Alexa fluor 488 and Alexa fluor 568, Molecular Probes). After washing with PBS, cells were stained with DAPI to stain nucleus. Each image was examined using a fluorescent microscope (EVOS, Invitrogen) or by flow cytometry. To exclude the integration of sendai virus, we performed qRT-PCR analysis with a specific probe and primers for a sendai viral gene (Sendai virus detection, SeV; TaqMan probe Mr04269880_mr; Sendai virus detection, SeV/Klf4; TaqMan probe Mr04421256_mr; GAPDH, VIC/ TAMRA probe 4310884E, Invitrogen). For qRT-PCR analysis, total RNA was extracted from each hiPSC line by using Direct-zol RNA purification Kit (Zymo research) and cDNA was synthesized with the ThermoScript™ RT- PCR system (Invitrogen). Four hiPSC lines from human male (two non-viral integrated clones)

and female (two non-viral integrated clones) wild type fibroblast cells were generated for this study. All the hiPSCs used in this study showed non-viral integration and a normal diploid karyotype.

**Proteomic analysis of human fibroblasts and hiPSCs**. BJ and PCS201 fibroblast cells, grown in DMEM + 10% FBS media, were dissociated by trypsin and collected by centrifugation. Human iPSCs (originated from BJ and PCS201 fibroblast cells) were grown in mTeSR media (STEMCELL Technologies) under Geltrex™ LDEV-Free hESC-qualified reduced growth factor basement membrane matrix (Thermo Fisher Scientific). Cells were dissociated by accutase (Thermo Fisher Scientific) and collected by centrifugation. Cells were lysed and their protein content digested into peptides by trypsin. After desalting, the peptides were labeled with TMT reagents. The labeled samples were equally mixed and further fractionated by neutral pH reverse phase liquid chromatography. In general, 10 fractions were collected and further analyzed by low pH reverse phase LC-MS/MS. During ion fragmentation, the TMT regents are cleaved to produce reporter ions for quantification. The collected data were searched against a database to identify peptides. While the peptides were identified by MS/MS, the quantification was achieved by the fragmented reporter ions in the same MS/MS scans. The peptide quantification data were then corrected for mixing errors and summarized to derive protein quantification results. Statistical analysis was performed to determine cutoff for altered proteins and to evaluate associated false discovery rate.

**HDAC2 binding**. The HDAC2 occupation data were downloaded from the Encyclopedia of DNA Elements (ENCODE) working group using the Jan 2011 data freeze (ftp://ftp.ebi.ac.uk/pub/databases/ensembl/encode/integration_data_jan2011). The optimal unified peak lists called with PeakSeq from two biological replicates from h1ESC, K562 leukemia and HEPG2 liver carcinoma cells present in Haib data sets were used (http://genome.ucsc.edu/cgi-bin/hgTrackUi?db=hg19&g=wgEncodeHaibTfbs). A gene was defined as bound by HDAC2 if a peak summit was within 5 kb of the transcript start sites. Fisher's exact test was calculated for this analysis.

**Motif analysis**. HDAC2-binding sites for all lysosomal genes or autophagy genes from all three cell lines downloaded from the ENCODE/Haib data set (http:// genome.ucsc.edu/cgi-bin/hgTrackUi?db=hg19&g=wgEncodeHaibTfbs) were combined. Motif search was performed using a 300-bp sequence near each peak summit, using the program HOMER.

**ChiP and gene expression analyses of available datasets**. For ChIP-seq dataset, peak results from MACS2 for U2OS MYC ChIP-seq data (GSE64425)[29] were associated with transcription start sites (TSS) of genes from the hg19 build of the human genome. Transcripts with peaks 5 kb up or down stream of the TSS were designated as MYC-bound in this cell line. Lists of genes associated with autophagy and lysosomes were then tested for enrichment among bound sites by Fisher's exact test. Peak results for medulloblastoma cells ChIP-seq data were obtained from GSE64425[40]. Data were remapped to hg19 with bwa/0.7.12, and coverage files were imported into the integrated genome viewer (IGV Broad, MIT), thereby documenting the binding of MYC to lysosomal genes and MiT/TFE members. Peak calling of ChIP-seq datasets performed in mouse group 3 medulloblastoma tumorspheres lacking Trp53 and overexpressing $Myc$ (control: input samples) was performed with MACS v.1.4.2 (the keep-dup parameter was adjusted, depending on the ChIP enrichment at the highest peaks)[40]. For the public RNAseq and microarray analysis, microarray raw data files (.Cel) from breast cancer cells (GSE60124), BJ cells (GSE43010) and the count matrix for RNAseq from normal endothelial cells (GSE77108) were downloaded from the GEO data base (NCBI). Array data were rma summarized and log2FC were computed using Partek Genomics Suite 6.6 (St Louis, MO USA). For the RNAseq, log2cpm were computed from the matrix using the log(cpm + 0.5)/log(2) transformation and these values were used to calculate the logFC. Next, a list of lysosomal associated genes (from array annotation, Affymetrix, Mountainview, CA) were compiled, de-duplicated and matched to the data by gene symbol. For each dataset the logFC of the HDAC inhibitor vs DMSO control for the logFC for genes in the lysosomal list were statistically compared to the remaining logFC values (one tailed $t$ test) and plotted in CDF plots using STATA MP/14.1 (College Station, TX USA). For analyses of gene expression on U2OS overexpressing MYC and silenced for MYC, we queried microarray data from Walz et al.[29].

**FOXH1 correlation analysis**. Microarray data from GSE50206[51] were used for the correlation between autophagy genes and FOXH1. Spearman correlations were used to correlate the FOXH1 expression with the autophagy genes present in this set.

**Statistical analyses**. Statistical analyses were performed using GraphPad Prism. Quantitative data are presented as mean ± SD. For comparisons between two groups, Student's $t$-test (unpaired, two-tailed) was performed. Groups were considered different when $p < 0.05$. For all quantifications, a minimum of three independent experiments were performed. Measurements were taken from distinct samples. Number of replicates is noted in the figure legends.

**Reporting summary**. Further information on research design is available in the Nature Research Reporting Summary linked to this article.

## Data availability

Expression microarray data are available at the Gene Expression Omnibus repository under accession number GSE106175. The data that support the findings of this study are available from the corresponding author, Alessandra d'Azzo, sandra.dazzo@stjude.org, upon request.

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

## Acknowledgements

We are indebted to D. Peach for his stimulating inputs. We thank A. Ballabio for critical reading of the manuscript; S. Frase, L. Horner for the electron microscopy images and quantification; R. Ashmun for the FACS analyses; C. Qu for help with motif analyses; and F. Harwood for assistance with the transfection protocols and reagents. We extend our gratitude to L. Attisano for her gift of the flag-FOXH1 plasmid, R. Puertollano for providing the Tfeb/Tfe3 dKO MEFs, K. Freeman for the neuroblastoma cells, M. Kundu for the LC3-GFP plasmid. A.d'A. holds the Jewelers for Children Endowed Chair in Genetics and Gene Therapy. This work was supported, in part, by NIH grants GM104981, and CA021764 (A.d'A.), P01CA096832, and P30 CA02165 (M.F.R.), the Assisi Foundation of Memphis (A.d'A.), the American Lebanese Syrian Associated Charities (ALSAC) (A.d'A.), and by a grant from the Deutsche Forschungsgemeinschaft (Emmy Noether Research Group, WO 2108/1-1) (E.W.).

## Author contributions

I.A. designed, performed, and analyzed most of the experiments. D.v.d.V. performed all the RT-qPCR, western blot analyses and assembled the figures. E.W. performed computational analyses of ChIP datasets. D.F. performed computational analyses of ChIP datasets, gene expression and correlation analyses of publicly available datasets. G.N. performed, analyzed, and deposited the microarray data. H.T. scored and analyzed tissue microarrays and patient-derived xenografts. E.M. performed immunohistochemistry, western blot analyses and data acquisition. R.M. performed enzymatic activity assays and cultured primary skin fibroblasts. Y.C. performed immunofluorescence analyses and co-immunoprecipitation analyses. M.F.R. provided the mouse group 3 medulloblastoma cells and intellectual input. J.A.W. and L.E.F. performed western blots analyses. X.Q. purified in house antibodies. M-J.H. reprogrammed the dermal fibroblasts into hIPSCs and performed proteomic analyses. G.C.G. provided reagents, interpreted data, and edited the paper. A.d'A. designed experiments, analyzed data, and supervised the study. I.A. and A.d'A. conceived the project and wrote the manuscript.

## Additional information

**Competing interests:** The authors declare no competing interests.

