## [Peer Review File · Nature Communications]

Reviewers' comments:

Reviewer #1 (Remarks to the Author):

The major new concept reported in this manuscript is that MYC suppresses the expression of genes encoding proteins related lysosomes and autophagy by acting in a competitive manner with the TFEB, TFE3 and MITF transcription factors by recruiting HDACs to their promoters and the promoters of their target genes. This is an interesting model that could have significant implications in multiple areas of biology and medicine. However, there are multiple issues with the data that is presented in support of this model that significantly reduce confidence in its accuracy. The most critical issues are outlined below.

1. Evidence that levels of lysosomal proteins increase in parallel with abundance of their transcripts in SAHA treated cells is lacking. Only minimal, unquantified data is presented for NEU1 and LAMP1 in Figure S2. If there is no major change in levels and/or activities of lysosomal proteins, then it is hard to imagine that the transcriptional changes have any physiological significance.
2. The text states that LAMP1 levels are increased in SAHA treated cells and cites data provided in Figure S2f. However, the data in this figure is conflicting on this topic. LAMP1 levels actually decrease in the SAHA treated HeLa cells but appear to increase modestly in the other 2 cell lines examined. This mismatch between what is stated and what is shown needs to be corrected. Furthermore, the lack of generalizability between cell lines should be explained.
3. In Figure S2, why is there such a large difference between changes in NEU1 protein levels and the measurements of enzymatic activity in control versus SAHA treated cells? Does the mismatch imply that most of the extra NEU1 in the SAHA treated cells is inactive?
4. Why does NEU1 mRNA stand out from other lysosome-related genes for the extent to which it is regulated by SAHA? Can this be explained by MYC and/or TFEB binding to its promoter?
5. Is there evidence that lysosomes actually function better in SAHA treated cells? Although implied, it is not currently demonstrated that the changes in lysosome number, appearance or gene expression is accompanied by enhanced function.

6. In addition to examining steady state levels of LC3 lipidation in SAHA-treated cells (Figure 4 and 6), the authors should examine autophagic flux to distinguish between enhanced autophagy versus impaired lysosome function as a cause of the changes in LC3 lipidation.

7. The inverse relationship between MYC and levels of lysosome/autophagy genes in fibroblasts versus hiPSCs is intriguing but no evidence of causation is shown.

8. The diagram in Figure 9 implies a direct role for MYC in recruiting HDACs to promoters of MIT/TFE gene. However, data in Figure 1e indicates that SAHA treatment does not affect HDAC2 abundance at the promoters of NEU1, LAMP1, MCOLN1, TFEB and TFE3 even though data in 2d shows that SAHA decreases the MYC occupancy at these promoters. This data argues for something other than MYC as the major determinant of HDAC2 levels at these promoters. This apparent discrepancy should be explained and/or the conclusions modified to better fit the data.

9. Does CHIP-Seq show greater abundance of TFEB/TFE3/MITF at the promoters of target genes following SAHA treatment? This would be expected based on the proposed model but the data presented in Figure 3b and 3c shows variable effects (some increases and some decreases) that do not readily match expectations.

Reviewer #2 (Remarks to the Author):

In this manuscript, Annunziata et al present compelling evidence that a HDAC/MYC–MIT/TFE axis might represent an epigenetic rheostat that regulates two major cellular catabolic machineries: lysosomal biogenesis and autophagy genes. By using the pan-HDAC inhibitor, SAHA, the investigators found a decreased binding of c-MYC to the promoters of lysosomal and autophagy genes, which allows occupancy of these promoters by the MIT/TFE transcriptional members, TFEB and TFE3 and the recently discovered autophagy regulator FOXH1. Overall, these findings provide novel insights into cell reprogramming and differentiation, “stemness” and how cells, by regulating these catabolic machineries might respond to physiological and pathological cues. Furthermore, they have potential translational implications for the treatment of lysosomal storage diseases as well as in cancer therapy.

However, before acceptance, the investigators should address the following minor comments:

1. Have the authors explored whether similar changes are observed in cells in which c-MYC is knocked-down or in cells treated with bromodomain inhibitors that have been shown to inhibit c-MYC.
2. Their data suggest an involvement of HDAC2. Have the authors explored whether more selective inhibitors (class I selective inhibitors) or knocking down HDAC2 might induce similar changes?
3. In some of the figures in which tubulin is used as a control, there seems to be increased tubulin expression. Is this associated with increased tubulin acetylation? If yes...is HDAC6 (which has been also found in the nuclei) playing a role?

Reviewer #3 (Remarks to the Author):

The authors suggest that c-MYC and HDACs inhibit lysosomal and autophagic biogenesis by repressing expression of MIT/TFE, FOXH1, as well as lysosomal and autophagy genes. The authors' conviction is that exposure to the HDAC inhibitor SAHA, inhibits c-MYC binding to these genes and allows TFE3, TFE3 and/or FOXH1 to bind and induce gene transcription. Moreover, c-MYC suppression of lysosomal and autophagic genes is also evident in pluripotent stem cells and cancer. There are several major and minor issues that need to be addressed.

Major Issues:

1. Many experimental details are missing from figure legends and methods sections. This is important to enable data interpretation and ensure reproducibility by independent labs. This is a major issue that must be resolved. For example, microarray expression analysis was performed (Fig 1), yet the dose and time of HeLa treatment with SAHA is not provided. The growth state of the cells is also not provided, which is important to evaluate whether control cells are proliferating and SAHA-treated cells are growth arrested at the time of RNA harvest, as this would be comparing different growth states of the cells as well as drug treatment. This information is essential for data interpretation.
2. Several other groups have evaluated gene expression in response to HDAC inhibitors. It is surprising that activation of lysosomal genes were not previously reported; only repression of cell cycle genes, as stated by the authors (top of page 6), in which they reference a review article (ref 32). The authors need to interrogate the publicly available datasets of HDAC inhibitor-treated cells to determine whether their observation of HDACi inducing lysosomal genes is evident in these other datasets, or if unique to the cells evaluated in this manuscript (HeLa, RH30, Sy5y).

3. Why were HeLa, RH30, Sy5y cell lines used for these studies? This needs to be incorporated into the text of the manuscript. Rationale is missing.
4. Supp Fig 1d: why are human fibroblasts chosen for these studies? Statistical analysis of this data is missing from the legend. If not significant, then text must be altered, "Similarly, SAHA induced lysosomal gene expression in the human cancer cell lines RH30 (rhabdomyosarcoma) and Sy5y (neuroblastoma) and in skin primary fibroblasts (Supplementary Fig. 1b-d)." Without statistical analysis/support of this statement, conclusions are incorrectly reported.
5. Two controls are missing from all ChIP analyses: 1) IgG control antibody; 2) non-specific binding control assaying an E-box that is not expected to bind HDAC2 (e.g. Fig 1e, 4f) or MYC (e.g. Fig 2d, 2e, 5b) or TFEB/TFE3 (e.g. Fig 3b-d); these can be derived from ENCODE data or a review of the literature for ChIP-qPCR for these proteins. Without the IgG control, how were the ChIP-qPCR analyses performed? Specifically, how was the % of input (y-axis) derived? Which of these values are significant? Statistical analysis is required for all ChIP assays. Without this important statistical analysis, one cannot interpret whether binding is increased or decreased. Thus, data interpretation is presently flawed.
6. Fig 1e: The authors state: "Inhibition of HDACs did not alter the binding of HDAC2 because SAHA specifically inhibits HDAC activity rather than affecting protein levels (Fig. 1e)." This result leaves the mechanism of SAHA-induced protein induction at these genes unclear. The assumption is that local histone acetylation is increased, leading to elevated expression. It is important to fill this gap and show that histone acetylation is increased at these lysosomal genes and TFEB/TFE3 as shown in Fig 1e. Without this data the authors cannot state their concluding statement of this paragraph describing this data (bottom of page 5, top of page 6), "This indicates that HDACs epigenetically control the levels of the MiT/TFE transcription factors."
7. SuppFig4: What dose/time of SAHA was used for these western blots showing MYC down-regulation? What was the growth state of the cells?
8. Fig 2d, 2e: What dose/time of SAHA was used for these MYC ChIP assays. What was the growth state of the cells? See previous comment #5 regarding missing controls.
9. Fig 2b: 'Inp' in the lower track is from the p53 null cells? This is not defined in the legend and needs to be made clear to the reader, otherwise the relevance of the legend statement "p53 null cells were used as control" is not clear.
10. Fig 2d and 2e: Evidence to show that MYC bound to these promoters is inducing (MYC) and repressing (MiT/TFE) expression needs to be shown for this statement to be validated. No evidence is provided that MYC induces its own transcription, as stated by the authors "We also observed that MYC was bound to its own promoter (Fig. 2e), upregulating its own transcription, while repressing the transcription of the MiT/TFE family members." This is surprising because MYC usually participates in a well-established autosuppression mechanism, leading to the down-regulation of expression.
11. Fig 3a: The authors state MYC expression is strongly downregulated, yet it appears not to be statistically significant. $p < 0.05$ is stated in the legend, yet what was analyzed and by which method is not mentioned. Statistical analysis needs to be applied to all data figures and the statistical method

used needs to be provided. Data interpretation is dependent upon this statistical analysis and is presently not provided.

12. Fig 3b-d: It appears from Fig 3b-d that TFEB and TFE3 could bind to these promoters prior to SAHA treatment, therefore the conviction that SAHA induced down-regulation of MYC was required for TFEB and TFE3 occupancy of these promoter regions is not valid. Again this speaks to data interpretation. The conclusion that TFEB can bind to its own promoter (Fig 3b) is also questionable. These conclusions are only possible with statistical analysis of the authors' data. Also for this ChIP analysis, controls are missing; see #5 above.

13. Fig 3e: The authors state, "To further understand how MYC fine-tunes lysosomal biogenesis, we tested a set of mouse Group 3 (G3) medulloblastoma cell lines, overexpressing Myc 34,35" This is unclear; a set of G3 cell lines were analyzed, however the figure (3e) shows only one value for each gene. Were several lines analyzed and then the results pooled?

14. Fig 3e and 3f: What was being compared in these figures? G3 cells with high MYC and p53 null compared to G3 cells with low MYC and p53 null? Legend statement is "Trp53^{-/-} mouse G3 medulloblastoma tumorspheres overexpressing Myc and Trp53^{-/-} cells were used as controls." This is unclear. Are these isogenic cell lines? The entire G3 cell system needs to be described and justified early in the manuscript. As the references related to G3 cells in this manuscript refer to primary human tumor analysis it was surprising to learn in the legend of Supp Fig 4 that the G3 cells were derived from a mouse model. There is no section in the methods on this G3 cell system and/or mouse model.

15. Supp 4a: Legend says HeLa, Figure says RH30; which is it?

16. Supp 4g: Apparent MYC peak needs to be highlighted to help readers understand authors' peak calls that they claim as significant. This is not clear. Statistical analysis not provided.

17. Supp 4j: Authors state in the legend "n=9; p <0.001", yet the expression of 4 genes are being analyzed and only one appears significant. This is not clear.

18. Fig 6 and Supp Fig 6: FOXH1 mRNA is shown but protein is not. Show that protein expression is also induced in response to SAHA. What concentration/time of SAHA was used? What was the growth state of the cells?

19. The authors state: "In addition, by querying MYC ChIP datasets from U2OS osteosarcoma cells 30, we found that MYC bound the promoter region of FOXH1, hence limiting its transcription (Fig. 6f)." Just because MYC is binding, based on ENCODE data, does not mean that it is repressing gene expression. ENCODE analysis is of U2OS cells, yet authors don't use U2OS cells and state that FOXH1 activity is cell-type dependent. The conclusions from this section on FOXH1 are not clear and have not been adequately validated.

20. Fig 6 legend: The authors state: "For immunoblots, C=anti-tubulin. G3= Trp53^{-/-} Mouse G3 medulloblastoma tumorspheres overexpressing Myc 34." It is hard to know what this statement refers to as it is associated with 6h, which is a heatmap.

21. Figure 7a: What is the mechanism of SAHA induction of NEU1-mutant mRNA gene expression in Fig 7a. Is local histone acetylation increased? A previous report suggest that KAT activity is not increased in response to HDACi, so this remains unclear and must be investigated in this system. (e.g. PMID:26380582)
22. Neither Fig 7a, 7b legend nor text explains 'Type 1 and 2' which are in the figures. Which patient derived fibroblasts in 7a correspond to those in 7b? Why is one of the fibroblasts missing in Fig 7a?
23. Fig 8a and Supp Fig 8a-c: Quantification is missing; how many patient samples were analyzed and what was the distribution of expression/localization. Statistical analysis is required.
24. Fig 8b-d: Examples of the western blots must be shown to accompany the results summarized in Fig 8b-d. These could be included as supplementary data.
25. Fig 9: Model suggests histone acetylation regulates autophagy and lysosomal genes, yet no evidence is provided to support this statement.

Minor Issues:

1. Fig 1e: Histogram fill is indistinguishable for SAHA and DMSO mock; what is DMSO mock and SAHA mock; these are not described in either the figure legend or materials and methods.
2. Fig 5d: Odd display of western blot; the two blots are shown as though they were two independent analyses of LC3 and tubulin. If so, then tubulin is not functioning as a loading control. If not, then classic presentation of these immunoblots should be displayed to show that the extracts resolved on one gel were transferred and then probed for the two proteins of interest.

Rebuttal to Reviewers' comments:

Reviewer #1

The major new concept reported in this manuscript is that MYC suppresses the expression of genes encoding proteins related lysosomes and autophagy by acting in a competitive manner with the TFEB, TFE3 and MITF transcription factors by recruiting HDACs to their promoters and the promoters of their target genes. This is an interesting model that could have significant implications in multiple areas of biology and medicine. However, there are multiple issues with the data that is presented in support of this model that significantly reduce confidence in its accuracy. The most critical issues are outlined below.

1. Evidence that levels of lysosomal proteins increase in parallel with abundance of their transcripts in SAHA treated cells is lacking. Only minimal, unquantified data is presented for NEU1 and LAMP1 in Figure S2. If there is no major change in levels and/or activities of lysosomal proteins, then it is hard to imagine that the transcriptional changes have any physiological significance.

R. Following the reviewer's suggestion we have tested the activities of several other lysosomal enzymes. The results of these assays show a general increase of lysosomal enzyme activities upon SAHA treatment. The data are now incorporated in Fig. 1e-i.

2. The text states that LAMP1 levels are increased in SAHA treated cells and cites data provided in Figure S2f. However, the data in this figure is conflicting on this topic. LAMP1 levels actually decrease in the SAHA treated HeLa cells but appear to increase modestly in the other 2 cell lines examined. This mismatch between what is stated and what is shown needs to be corrected. Furthermore, the lack of generalizability between cell lines should be explained.

R: We have now quantified the levels of LAMP1, which is generally increased in all SAHA treated cells. Results from these analyses are found in Supplementary Fig. 2i-k. As to be foreseen, the effects of SAHA treatment differ in amplitude in tumor cells of different derivation; however, upon SAHA treatment all cells show consistent and reproducible increases of all lysosomal constituents tested.

3. In Figure S2, why is there such a large difference between changes in NEU1 protein levels and the measurements of enzymatic activity in control versus SAHA treated cells? Does the mismatch imply that most of the extra NEU1 in the SAHA treated cells is inactive?

R: We completely agree with the reviewer on this point. We reason that this discrepancy could be at least in part due to the limitations of using a synthetic substrate to assay NEU1 enzyme activity. In addition, given the dependency of NEU1 on its interaction with the protective protein/cathepsin A (PPCA) for its lysosomal stability and catalytic activation. It is conceivable, as the reviewer states, that some of the extra NEU1 in SAHA treated cells is not catalytically active, because SAHA treatment increases PPCA levels to a lesser extent than those of NEU1. Further studies are in progress to explain the

strong and seemingly unique response of NEU1 to HDAC inhibition, but this analysis goes beyond the scope of the present manuscript and will be part of a follow-up manuscript.

4. Why does NEU1 mRNA stand out from other lysosome-related genes for the extent to which it is regulated by SAHA? Can this be explained by MYC and/or TFE3 binding to its promoter?

R: Again, we fully agree with the reviewer on this interesting point. We think that it is due to the binding of TFE3, which avidly occupies the promoter of NEU1 (Fig. 3e).

5. Is there evidence that lysosomes actually function better in SAHA treated cells? Although implied, it is not currently demonstrated that the changes in lysosome number, appearance or gene expression is accompanied by enhanced function.

R: The reviewer raises a question that at the moment can only be answered by deductive but indirect observations. Because lysosomes, as other organelles, are very heterogeneous, it is a difficult question to address. The commonly used approach is to test the activity of hydrolytic enzymes and their function towards known substrates. We have now included data showing that activities of many other lysosomal enzymes are increased in SAHA treated cells. We have demonstrated an expansion of the lysosomal system in SAHA treated cells that is paralleled by an increased number of lysosomes, and upregulation of transcription factors regulating lysosomal biogenesis and function. From these combined results we can infer that the overall function of the lysosomal system is increased in SAHA treated cells.

6. In addition to examining steady state levels of LC3 lipidation in SAHA-treated cells (Figure 4 and 6), the authors should examine autophagic flux to distinguish between enhanced autophagy versus impaired lysosome function as a cause of the changes in LC3 lipidation.

R: Following the suggestion of the reviewer we have now assayed autophagic flux using a tandem construct carrying EGFP-mCherry-LC3 according to the method described by Dupont et al 2014 Methods in Enzymology Volume 543, pages 73-88. The results are now incorporated in Fig. 4f,g.

7. The inverse relationship between MYC and levels of lysosome/autophagy genes in fibroblasts versus hiPSCs is intriguing but no evidence of causation is shown.

R: We fully agree with the reviewer that these results are intriguing. We have included them in the manuscript as an example of the need for modulating the lysosomal autophagic systems in reprogrammed cells vs differentiated cells. However, dissecting the cause/effect of this inverse relationship is outside the scope of this paper. This study is part of an ongoing investigation done in collaboration with Dr. Han and will be the subject of a follow-up manuscript. Dr. Han, who is an expert of hiPSCs is specifically interested in comparing the metabolic state of fibroblasts vs fibroblast-derived hiPSCs (Cha Y, Han MJ et al. Nature Cell Biology 2017).

8. The diagram in Figure 9 implies a direct role for MYC in recruiting HDACs to promoters of MIT/TFE gene. However, data in Figure 1e indicates that SAHA treatment does not affect HDAC2 abundance at the promoters of NEU1, LAMP1, MCOLN1, TFEB and TFE3 even though data in 2d shows that SAHA decreases the MYC occupancy at these promoters. This data argues for something other than MYC as the major determinant of HDAC2 levels at these promoters. This apparent discrepancy should be explained, and/or the conclusions modified to better fit the data.

R. The apparent discrepancy the reviewer is referring to is due to the fact that SAHA affects only the catalytic activity of HDACs. Crystallographic studies have shown that SAHA inhibits HDAC activity by binding to the active site of the enzyme. Specifically, the hydroxamic end of the SAHA molecule binds to the zinc atom within the catalytic pocket of HDACs, with the phenyl ring of SAHA projecting out towards the surface of these enzymes, blocking their activity without altering their protein levels. Upon SAHA treatment, HDACs, while still occupying the promoters of NEU1, LAMP1, MCOLN1, TFEB and TFE3 (used as examples) (Fig. 1p), can no longer function catalytically. Blocking the activity of HDACs with SAHA or directly silencing HDAC2 (see new Fig. 2b) affects the levels of MYC (Fig. 2b and Supplementary Fig. 4a-c). Consequently, lowering the concentration of MYC results in its displacement from the promoters of NEU1, LAMP1, MCOLN1, TFEB and TFE3 (Fig. 2h,i), that now are being occupied by TFEB and TFE3, thus promoting transcriptional activation of lysosomal genes.

9. Does CHIP-Seq show greater abundance of TFEB/TFE3/MITF at the promoters of target genes following SAHA treatment? This would be expected based on the proposed model but the data presented in Figure 3b and 3c shows variable effects (some increases and some decreases) that do not readily match expectations.

R: Following the reviewer's concerns, we have now repeated and quantified all the CHIP assays. The new results are incorporated in Fig. 3c-e.

Reviewer #2

In this manuscript, Annunziata et al present compelling evidence that a HDAC/MYC-MIT/TFE axis might represent an epigenetic rheostat that regulates two major cellular catabolic machineries: lysosomal biogenesis and autophagy genes. By using the pan-HDAC inhibitor, SAHA, the investigators found a decreased binding of c-MYC to the promoters of lysosomal and autophagy genes, which allows occupancy of these promoters by the MIT/TFE transcriptional members, TFEB and TFE3 and the recently discovered autophagy regulator FOXH1. Overall, these findings provide novel insights into cell reprogramming and differentiation, "stemness" and how cells, by regulating these catabolic machineries might respond to physiological and pathological cues. Furthermore, they have potential translational implications for the treatment of lysosomal storage diseases as well as in cancer therapy.

However, before acceptance, the investigators should address the following minor comments:

1. Have the authors explored whether similar changes are observed in cells in which c-MYC is knocked-down or in cells treated with bromodomain inhibitors that have been shown to inhibit c-MYC.

R: To address this very pertinent point raised by the reviewer, we have now included data obtained from cells in which MYC was silenced by siRNA (Walz et al 2014). In these cells we confirm the occurrence of transcriptional activation of the lysosomal and autophagy genes as well as of the MIT/TFE genes. Results from these analyses are presented in Fig. 3h, i, and Fig. 5f.

2. Their data suggest an involvement of HDAC2. Have the authors explored whether more selective inhibitors (class I selective inhibitors) or knocking down HDAC2 might induce similar changes?

R: As the reviewer pinpoints, by using a pan HDAC inhibitor, we cannot exclude that other HDACs, besides HDAC2, regulate lysosomal biogenesis. We have chosen HDAC2 as a proxy for our analyses because it is one of the most abundant HDACs, for which there are very solid CHIP data in the ENCODE database. However, to address the reviewer's comment, we have performed additional experiments using the class I selective inhibitor Romidepsin, a potent natural prodrug inhibitor of HDAC1 and HDAC2 (IC₅₀ values of 36 nM for HDAC1 and 47 nM for HDAC2). Additionally, we have silenced HDAC2 in HeLa cells. In agreement with our results with SAHA, treatment with Romidepsin or silencing of HDAC2 resulted in increased activity of lysosomal enzymes (Fig. 1j-n). Furthermore, the silencing of HDAC2 (Supplementary Fig. 3e-k) decreased the levels of MYC, a finding that points to a direct role of this HDAC in regulating MYC (Fig. 2b,c).

3. In some of the figures in which tubulin is used as a control, there seems to be increased tubulin expression. Is this associated with increased tubulin acetylation? If yes...is HDAC6 (which has been also found in the nuclei) playing a role?

R: Following the reviewer's suggestion, we have now performed Western blots using acetylated tubulin antibody in SAHA treated cells. The reviewer's comment was indeed correct, because the results of this analysis confirm an increase in tubulin acetylation. For this reason, we have now used Coomassie staining as loading control for all Western blots (Welinder et al J. Proteome Research 2011) and repeated the quantification of the relevant bands. - The reviewer raises an interesting point regarding HDAC6. To address this comment, we have performed real time PCR analyses of the class I HDACs and HDAC6 in HeLa, RH30 and Sy5y. The results, shown below, demonstrate the preferential expression of HDAC1 and HDAC2, while HDAC6 is expressed at very low levels. For this reason, we think it is unlikely that HDAC6 is involved in the epigenetic regulation of lysosomal and autophagic genes.

Reviewer #3 (Remarks to the Author):

The authors suggest that c-MYC and HDACs inhibit lysosomal and autophagic biogenesis by repressing expression of MIT/TFE, FOXH1, as well as lysosomal and autophagy genes. The authors' conviction is that exposure to the HDAC inhibitor SAHA, inhibits c-MYC binding to these genes and allows TFE3, TFE3 and/or FOXH1 to bind and induce gene transcription. Moreover, c-MYC suppression of lysosomal and autophagic genes is also evident in pluripotent stem cells and cancer. There are several major and minor issues that need to be addressed.

Major Issues:

1. Many experimental details are missing from figure legends and methods sections. This is important to enable data interpretation and ensure reproducibility by independent labs. This is a major issue that must be resolved. For example, microarray expression analysis was performed (Fig 1), yet the dose and time of HeLa treatment with SAHA is not provided. The growth state of the cells is also not provided, which is important to evaluate whether control cells are proliferating and SAHA-treated cells are growth arrested at the time of RNA harvest, as this would be comparing different growth states of the cells as well as drug treatment. This information is essential for data interpretation.

R: We respectfully point out to the reviewer that information about the dose of SAHA and time of treatment was detailed in the Methods. However, we have now specified the dose of SAHA and time of treatment (20 μ M for 24 hr is the regimen used throughout the studies) in all legends to the Figures (Fig.1, Fig.2, Fig.3, Fig.4, Fig.5 Fig.6, Fig.7 and Supplementary Fig.1, Fig.2, Fig.3, Fig.4, Fig.5, Fig.6, Fig.7). In addition, the protocol for SAHA treatment was carefully designed based on numerous published studies (for example: Grabarska et al J Cancer 2017; Medvedev et al Science Advances 2018; Yamamoto et al Anticancer Research 2008; Sonnemann et al BMC Cancer 2006; Stamatopoulos et al Haematologica 2010; McGee-Lawrence et al Bone 2011), and we have used the exact same type of reported controls. To address this comment of the reviewer, we now provide evidence that silencing of HDAC2 (Fig. 2b,c and Supplementary Fig. 3e-k) has similar effects on lysosomal biogenesis as SAHA or Romidepsin treatment. Romidepsin is an additional HDAC inhibitor that we have now tested to corroborate the results obtained with SAHA (Fig. 1j-n; Fig. 3b; Supplementary Fig. 3d).

2. Several other groups have evaluated gene expression in response to HDAC inhibitors. It is surprising that activation of lysosomal genes were not previously reported; only repression of cell cycle genes, as stated by the authors (top of page 6), in which they reference a review article (ref 32). The authors need to interrogate the publicly available datasets of HDAC inhibitor-treated cells to determine whether their observation of HDACi inducing lysosomal genes is evident in these other datasets, or if unique to the cells evaluated in this manuscript (HeLa, RH30, Sy5y).

R: As the reviewer states, it is indeed surprising that other groups have not previously identified changes in the lysosomal system in response to HDAC inhibitors. That is the reason why this represents one of the novel findings of our paper. As requested by the

reviewer, besides the several cell types tested here, we have now also interrogated publicly available datasets (GSE77108, GSE43010, GSE60124) and found a similar activation of the lysosomal system. These analyses are now included in Supplementary Fig. 1e-g.

3. Why were HeLa, RH30, Sy5y cell lines used for these studies? This needs to be incorporated into the text of the manuscript. Rationale is missing.

R: Because many of the studies published so far, which focused on lysosomal biogenesis were performed nearly exclusively in HeLa cells, we purposely expanded the scope of the study to other tumor cells and primary cells to demonstrate the generality of our results. For this reason, beside HeLa cells, we have included in the study RH30, a rhabdomyosarcoma cell line of mesenchymal origin; Sy5y, a neuroblastoma cell line; and human primary fibroblasts. We have amended the text explaining the use of the different cell lines in the revised manuscript.

4. Supp Fig 1d: why are human fibroblasts chosen for these studies? Statistical analysis of this data is missing from the legend. If not significant, then text must be altered, "Similarly, SAHA induced lysosomal gene expression in the human cancer cell lines RH30 (rhabdomyosarcoma) and Sy5y (neuroblastoma) and in skin primary fibroblasts (Supplementary Fig. 1b-d)." Without statistical analysis/support of this statement, conclusions are incorrectly reported.

R: We think it is appropriate to also test primary cells such as skin fibroblasts, and not focus our studies on cancer cell lines only. This generalizes our findings, which clearly are not restricted to cancer cell lines. Statistical analysis is now added to the legend of Supplementary Fig. 1d.

5. Two controls are missing from all ChIP analyses: 1) IgG control antibody; 2) non-specific binding control assaying an E-box that is not expected to bind HDAC2 (e.g. Fig 1e, 4f) or MYC (e.g. Fig 2d, 2e, 5b) or TFEB/TFE3 (e.g. Fig 3b-d); these can be derived from ENCODE data or a review of the literature for ChIP-qPCR for these proteins. Without the IgG control, how were the ChIP-qPCR analyses performed? Specifically, how was the % of input (y-axis) derived? Which of these values are significant? Statistical analysis is required for all ChIP assays. Without this important statistical analysis, one cannot interpret whether binding is increased or decreased. Thus, data interpretation is presently flawed.

R: 1) As originally specified in the Methods, we have used normal IgG antibody a control in all ChIP assays. They were labelled in the figures as "mock ". To avoid confusion, we have now labelled the control bars in all figures (Figs. 1p, q; 2h, i; 3c-e; 4i, j; 5b; 7d) as "IgG" and clarified this point in the Methods. 2) For non-specific binding, we have chosen a NEU1 genomic region at +5 kb from the TSS of NEU1 that showed no binding to HDAC2, MYC, TFEB and TFE3. This non-specific control is now indicated as NEU1 INT (internal) in Fig. 1p, q; Fig. 2h; Fig. 3c, e. 3). The quantification for all ChIP assays was calculated as % of input. By this calculation mode, the ChIP signals were divided by the signals obtained from the input sample and normalized for both

background levels and input. Results were calculated as $100 \times 2^{\Delta Ct}$ (Adjusted input - Ct (IP)). Statistical analyses for all ChIP assays are now included.

6. Fig 1e: The authors state: "Inhibition of HDACs did not alter the binding of HDAC2 because SAHA specifically inhibits HDAC activity rather than affecting protein levels (Fig. 1e)." This result leaves the mechanism of SAHA-induced protein induction at these genes unclear. The assumption is that local histone acetylation is increased, leading to elevated expression. It is important to fill this gap and show that histone acetylation is increased at these lysosomal genes and TFEB/TFE3 as shown in Fig 1e. Without this data the authors cannot state their concluding statement of this paragraph describing this data (bottom of page 5, top of page 6), "This indicates that HDACs epigenetically control the levels of the MiT/TFE transcription factors."

R: To address the reviewer's concerns we have now performed ChIP with acetylated histone antibody, followed by real time PCR of the promoters of lysosomal genes and TFEB/TFE3 genes. The results clearly show induction of acetylated lysine 14 on histone H3. These data are now included in Fig. 1q.

7. SuppFig4: What dose/time of SAHA was used for these western blots showing MYC down-regulation? What was the growth state of the cells?

R: Please see our reply to point #1.

8. Fig 2d, 2e: What dose/time of SAHA was used for these MYC ChIP assays. What was the growth state of the cells? See previous comment #5 regarding missing controls.

R: Please see our reply to point #1.

9. Fig 2b: 'Inp' in the lower track is from the p53 null cells? This is not defined in the legend and needs to be made clear to the reader, otherwise the relevance of the legend statement "p53 null cells were used as control" is not clear.

R: We thank the reviewer for pinpointing this potentially confusing statement. We have now amended the legend to Fig. 2e-g (previously Fig. 2b).

10. Fig 2d and 2e: Evidence to show that MYC bound to these promoters is inducing (MYC) and repressing (MiT/TFE) expression needs to be shown for this statement to be validated. No evidence is provided that MYC induces its own transcription, as stated by the authors "We also observed that MYC was bound to its own promoter (Fig. 2e), upregulating its own transcription, while repressing the transcription of the MiT/TFE family members." This is surprising because MYC usually participates in a well-established autosuppression mechanism, leading to the down-regulation of expression.

R: We agree with the reviewer on this point and acknowledge that based on our results we cannot conclude that MYC induces its own transcription. We have now modified the text to properly describe the results on MYC expression.

11. Fig 3a: The authors state MYC expression is strongly downregulated, yet it appears

not to be statistically significant. $p < 0.05$ is stated in the legend, yet what was analyzed and by which method is not mentioned. Statistical analysis needs to be applied to all data figures and the statistical method used needs to be provided. Data interpretation is dependent upon this statistical analysis and is presently not provided.

R: We have modified and edited the legend to Fig. 3a, which now includes statistical significance values for the different transcription factors as well as asterisks to define statistical significance.

12. Fig 3b-d: It appears from Fig 3b-d that TFEB and TFE3 could bind to these promoters prior to SAHA treatment, therefore the conviction that SAHA induced down-regulation of MYC was required for TFEB and TFE3 occupancy of these promoter regions is not valid. Again this speaks to data interpretation. The conclusion that TFEB can bind to its own promoter (Fig 3b) is also questionable. These conclusions are only possible with statistical analysis of the authors' data. Also for this ChIP analysis, controls are missing; see #5 above.

R: As mentioned before, statistical analyses for the ChIP assays are now included (Fig. 1p,q; Fig. 2h,i; Fig. 3c-e; Fig. 4i,j; Fig. 5b; Fig. 7d). We want to reiterate that the proposed mechanism entails the dynamic relationship between MYC and TFEB/TFE3 transcription factors. Our model is clearly dependent on the concentration of these transcription factors. Of course, under normal conditions a certain amount of TFEB/TFE3 must occupy the promoters of lysosomal genes to allow for their transcription. We have used SAHA treatment in these studies to modulate the concentration of these transcription factors in order to prove their mutual exclusivity.

13. Fig 3e: The authors state, "To further understand how MYC fine-tunes lysosomal biogenesis, we tested a set of mouse Group 3 (G3) medulloblastoma cell lines, overexpressing Myc 34,35" This is unclear; a set of G3 cell lines were analyzed, however the figure (3e) shows only one value for each gene. Were several lines analyzed and then the results pooled?

R: The medulloblastoma cell lines used in these experiments were generated and characterized in the lab of Dr. Roussel and described in Kawauchi *et al.* Cancer Cell 2012 and Vo *et al.* Cancer Cell 2016. Individual neurosphere cell lines numbered 19251; 19554, 15486; 13465; 19568, derived from mouse medulloblastoma tumors, a model of human group 3 medulloblastoma, were assayed separately and the obtained values were averaged. Both material and methodology are now detailed in the Methods section of the revised paper and in the legend to Fig. 2 e-g; Fig. 3f,g; Fig. 5a,b; Fig. 5c-e; Fig. 6j and Supplementary Fig. 4h-l; Fig. 6j.

14. Fig 3e and 3f: What was being compared in these figures? G3 cells with high MYC and p53 null compared to G3 cells with low MYC and p53 null? Legend statement is "Trp53^{-/-} mouse G3 medulloblastoma tumorspheres overexpressing Myc and Trp53^{-/-} cells were used as controls." This is unclear. Are these isogenic cell lines? The entire G3 cell system needs to be described and justified early in the manuscript. As the references related to G3 cells in this manuscript refer to primary human tumor analysis it was surprising to learn in the legend of Supp Fig 4 that the G3 cells were derived from a

mouse model. There is no section in the methods on this G3 cell system and/or mouse model.

R: We respectfully correct the reviewer on this point. As mentioned above, the medulloblastoma cell lines were derived from mouse medulloblastoma (Kawauchi *et al* Cancer Cell 2012 and Vo *et al* Cancer Cell 2016). The original submission of the manuscript included the latter references in the Methods. We have now amended this paragraph adding more details about the neurosphere cell lines and modified the Figure legends accordingly. The neurosphere cultures are indeed isogenic and overexpressed MYC in a Trp53^{-/-} background (lines 19251; 19554, 15486; 13465; 19568); they were compared to Trp53^{-/-} cells with no MYC overexpression. We have corrected the text referring to this point as well as the legends to Fig. 2 e-g; Fig. 3f,g; Fig. 5a,b; Fig. 5c-e; Fig. 6j and to Supplementary Fig. 4h-l; Fig. 6j.

15. Supp 4a: Legends says HeLa, Figure says RH30; which is it?

R: We thank the reviewer for pointing out this discrepancy. Supplementary Fig. 4e (original submission Supplementary Fig. 4a) indeed refers to RH30. We have now corrected the legend.

16. Supp 4g: Apparent MYC peak needs to be highlighted to help readers understand authors' peak calls that they claim as significant. This is not clear. Statistical analysis not provided.

R: For MYC binding (Supplementary Fig. 4d), we have visualized the raw data (tags in the ChIP-seq) at the promoters of the different genes in comparison to input; the results are shown as normalized tag counts. Peaks have been highlighted by dotted-line rectangles.

17. Supp 4j: Authors state in the legend "n=9; p <0.001", yet the expression of 4 genes are being analyzed and only one appears significant. This is not clear.

R: We have now added asterisks in this figure (Supplementary Fig. 4e-g) to define statistical significance and modified the legend.

18. Fig 6 and Supp Fig 6: FOXH1 mRNA is shown but protein is not. Show that protein expression is also induced in response to SAHA. What concentration/time of SAHA was used? What was the growth state of the cells?

R: Please see our reply to point #1. As requested by the reviewer, protein expression for FOXH1 after SAHA treatment is now shown in Fig. 6f.

19. The authors state: "In addition, by querying MYC ChIP datasets from U2OS osteosarcoma cells 30, we found that MYC bound the promoter region of FOXH1, hence limiting its transcription (Fig. 6f)." Just because MYC is binding, based on ENCODE data, does not mean that it is repressing gene expression. ENCODE analysis is of U2OS cells, yet authors don't use U2OS cells and state that FOXH1 activity is cell-type

dependent. The conclusions from this section on FOXH1 are not clear and have not been adequately validated.

R: It is well established that MYC binds to many promoters. In response to the reviewer's comment we have now added FOXH1 expression data derived from Walz et al., 2014 that show FOXH1 repression when MYC is overexpressed in U2OS cells (thus, bound to the FOXH1 promoter) and FOXH1 induction when MYC is silenced in the same cell line. Results from these analyses are now included in Fig. 6i, k.

20. Fig 6 legend: The authors state: "For immunoblots, C=anti-tubulin. G3= Trp53-/- Mouse G3 medulloblastoma tumorspheres overexpressing Myc 34." It is hard to know what this statement refers to as it is associated with 6h, which is a heatmap.

R: We have modified the legend to Fig. 6j to clarify this point.

21. Figure 7a: What is the mechanism of SAHA induction of NEU1-mutant mRNA gene expression in Fig 7a. Is local histone acetylation increased? A previous report suggest that KAT activity is not increased in response to HDACi, so this remains unclear and must be investigated in this system. (e.g. PMID:26380582)

R: To address the reviewer's concern, we have now performed ChIP with acetylated histone H3 K14 antibody followed by real time PCR of the NEU1 promoter using fibroblasts from one of the patients with sialidosis. Results of this experiment are shown in Fig. 7d.

22. Neither Fig 7a, 7b legend nor text explains 'Type 1 and 2' which are in the figures. Which patient derived fibroblasts in 7a correspond to those in 7b? Why is one of the fibroblasts missing in Fig 7a?

R: We apologize for omitting this detail in the text. Type I and type II refer to the two clinical forms of sialidosis, type I being less severe than type II. We have now added this description to the text and in the legend to Fig. 7. We have also included the expression analysis of NEU1 after SAHA treatment in fibroblasts from a second patient with type II sialidosis, now shown in Fig. 7a.

23. Fig 8a and Supp Fig 8a-c: Quantification is missing; how many patient samples were analyzed and what was the distribution of expression/localization. Statistical analysis is required.

R: We have added the quantification and the distribution of expression/localization, as requested by the reviewer. The reason for showing exclusively the quantification of the nuclear staining in the case of the RH30r xenograft samples is because the cells have a very minute amount of cytoplasm that was impossible to correctly quantify. However, the results are in line with our proposed mechanism and are shown in Fig. 8b and Supplementary Fig. 8d, e.

24. Fig 8b-d: Examples of the western blots must be shown to accompany the results summarized in Fig 8b-d. These could be included as supplementary data.

R: As requested by the reviewer, we have now added several Western blots that support the data shown in Fig. 8 c-e. The immunoblots are included in Supplementary Fig.9 c-j.

25. Fig 9: Model suggests histone acetylation regulates autophagy and lysosomal genes, yet no evidence is provided to support this statement.

R: To address this comment, we have performed new ChIP analyses with acetylated histone antibody that are now shown in Fig. 1q, Fig. 4j and Fig. 7d. The data strongly support our proposed model.

Minor Issues:

1. Fig 1e: Histogram fill is indistinguishable for SAHA and DMSO mock; what is DMSO mock and SAHA mock; these are not described in either the figure legend or materials and methods.

R: We have changed the colors of the graphs in the figures to make it clearer to the reader.

2. Fig 5d: Odd display of western blot; the two blots are shown as though they were two independent analyses of LC3 and tubulin. If so, then tubulin is not functioning as a loading control. If not, then classic presentation of these immunoblots should be displayed to show that the extracts resolved on one gel were transferred and then probed for the two proteins of interest.

R: We have removed the word “medulloblastoma” and changed the Figure. The loading control is now in Supplementary Fig. 6i and refers to the blot that was probed with LC3 antibody, shown in Fig. 5d.

Reviewers' comments:

Reviewer #1 (Remarks to the Author):

This manuscript strings together together a wide range of experiments that lead the authors to conclude that MYC suppresses lysosome function and autophagy by interfering with the ability of MIT-TFE and FOXH1 transcription factors to regulate the expression of genes that encode proteins which function in these pathways. This appears to occur both through the regulation of TFEB and TFE3 expression as well as by competing for binding sites in the promoters of target genes and/or recruiting histone deacetylases (HDACs) to such promoters. However, close inspection of the data raises questions about whether the major conclusions are robustly justified. In particular, there is an over-reliance on interpretations that are based on correlations. In particular, although HDAC inhibition has effects on MYC, TFEB, TFE3, FOXH1, lysosomes and autophagy, a causal chain of events between them has not been rigorously established. As a result, there is considerable uncertainty about the model proposed in Figure 9 that implies a direct cause and effect relationship. For these reasons, there is a significant mismatch between the confidence with which a novel model is proposed and the actual data provided in support of this model.

1. It remains unproven that the changes in TFEB and TFE3 explain the changes in lysosome/autophagy gene expression and function in response to SAHA. Does SAHA still regulate this pathway in TFEB and TFE3 depleted cells or over-expressing cells?

2. Figures 1 and 2 suggest that TFEB and TFE3 are regulated in a similar way by SAHA. However, Figure 3a and 3b show that the abundance of their transcripts changes in opposite directions following treatment with SAHA.

3. Does SAHA still regulate the expression of lysosomal genes in cells that lack MYC?

4. It remains difficult to make a link between changes in gene expression and actual function of lysosomes. For example, NEU1 mRNA levels were found to increase ~16 fold in response to SAHA treatment. Figure S2 suggests that there are similar changes in NEU1 protein levels. Meanwhile, the NEU1 activity only increased ~2 fold (Fig. 1). Can this difference be explained? What happens to NEU1 abundance in lysosomes? Could there be a trafficking defect? The same is true for other lysosomal enzymes where the changes in mRNA levels far exceed changes in their activities. Some speculation on this topic was provided in the response to the reviewers. This should also be more clearly presented in the actual manuscript in order to help readers make sense of this data.

5. The autophagic flux assays in Figure 4f and g are puzzling as it appears that starvation has no effect on autophagic flux (this goes against the well established induction of autophagy in response to starvation). This raises questions about whether this assay is effectively measuring autophagic flux. A simpler alternative, would be to examine the ratio of lipidated to non-lipidated LC-3 by immunoblot analysis in cells treated +/-SAHA and +/-bafilomycin A. Given that the SAHA part of this experiment was already done in 4b, this is certainly feasible.

6. Fig. 1 D. Electron microscopy quantification refers to analysis of 20 images. How many biological replicates does this represent?

7. Why are error bars in the graph presented in 1q presented as SEM while other panels in this figure use SD?

8. Figure S1 is titled "Lysosomal genes are epigenetically regulated". In reality, all that is shown is that they are regulated by treatment with SAHA.

9. Figure S4 is titled "Supplementary Fig. 4. MYC levels are epigenetically regulated and control the expression of the MiT/TFE transcription factors." In reality, all that is shown is that SAHA treatment has effects on the abundance of MYC, TFEB, TFE3. No causal relationship is established by the data in this figure. In contrast, the titles of Supplemental Figures 5 and 6 are much more reasonable.

Reviewer #2 (Remarks to the Author):

In this revised manuscript, the authors have properly addressed all my critiques. In particular the experiments in which HDAC2 was genetically or pharmacologically disrupted, support the role of this particular HDAC in regulating in regulating MYC.

Overall, this revised version is clearly superior to its predecessor and as such this manuscript should be accepted for publication.

Reviewer #3 (Remarks to the Author):

The authors have addressed many of the issues raised by reviewers and the manuscript has been improved substantially. Two major issues remain, as well as a few minor points.

1. The authors conclude that MYC and HDAC interact at the target promoters to repress gene transcription (Fig 9), yet no evidence is provided that MYC and HDACs interact or are even co-bound to the target promoters. Additional evidence, for MYC:HDAC interaction such as proximity ligation assay and/or co-ip are required. In addition, to provide evidence that MYC:HDAC co-regulate target gene transcription ChIP/Re-ChiP showing the MYC ChIP material could then be used to ChIP HDACs would provide experimental support for their model. This is important as this model of MYC:HDAC repression has not been substantiated by others. The authors only cite one BBRC paper from 2004 to support their claim (ref 29).

2. The mechanism of MYC down-regulation in response to HDAC inhibition remains unclear. The authors suggest that HDACs are directly regulating MYC gene transcription and that inhibition of HDACs directly causes the down-regulation of MYC expression. No evidence is provided to show that HDACs are bound to the MYC promoter, per se. Also, one presumes HDACs at the MYC promoter would suppress MYC expression and inhibiting HDACs would lead to the up-regulation of MYC, yet MYC is down-regulated in response to HDAC inhibition. This is a major point of confusion that the authors need to clarify.

It is likely that the down-regulation of MYC is an indirect consequence of HDAC inhibition driving cellular growth arrest/autophagy, which in turn leads to the down-regulation of MYC expression. The authors did not respond to my query regarding the growth state of the cells, which was raised three times in the original review. Perhaps the authors don't understand why this issue is important. The issue is 'cause or consequence'. For example, MYC protein levels decrease in response to shRNAs targeting HDAC2 (Fig2b). Is this because HDACs regulate MYC transcription, as the authors suggest, or because knockdown of HDACs leads to decreased cell proliferation and therefore MYC is downregulated. Similarly, in response to SAHA treatment, MYC no longer ChIPs to promoters (Fig 2h, i). Is this because SAHA decreases MYC protein levels due to a direct (HDAC regulation of MYC expression; a causal effect) or indirect (SAHA triggered growth arrest, leading to MYC down-regulation; a consequential effect)? This cause or consequence issue repeats throughout the manuscript, e.g. Fig 3a. It would be incorrect to leave the reader thinking that inhibiting HDACs directly down-regulates MYC expression. This needs clarification either through additional experimentation or by including this caveat in the text accordingly.

Minor issues:

1. Line 152: 'important noticing' should be 'important to notice'

2. Figures: The controls for a given experiment need to be reported as part of that experiment and not as separate figure panels. By displaying as independent panels, there is no confidence that the controls were run at the same time as the experiment. For example, Fig 2b and 2c; supp fig 4a and 4b, etc. These need to be changed, e.g. Fig 2b data (left panel), controls (right panel).

3. Review article references are very old (from the 1990's) and need to be updated, e.g. Ref 26 and 27

Response to Reviewers' comments:

Reviewer #1 (Remarks to the Author):

This manuscript strings together a wide range of experiments that lead the authors to conclude that MYC suppresses lysosome function and autophagy by interfering with the ability of MIT-TFE and FOXH1 transcription factors to regulate the expression of genes that encode proteins which function in these pathways. This appears to occur both through the regulation of TFEB and TFE3 expression as well as by competing for binding sites in the promoters of target genes and/or recruiting histone deacetylases (HDACs) to such promoters. However, close inspection of the data raises questions about whether the major conclusions are robustly justified. In particular, there is an over-reliance on interpretations that are based on correlations. In particular, although HDAC inhibition has effects on MYC, TFEB, TFE3, FOXH1, lysosomes and autophagy, a causal chain of events between them has not been rigorously established. As a result, there is considerable uncertainty about the model proposed in Figure 9 that implies a direct cause and effect relationship. For these reasons, there is a significant mismatch between the confidence with which a novel model is proposed and the actual data provided in support of this model.

R: We have successfully addressed all additional concerns raised by this reviewer. All new data are presented in 19 new panels that we have incorporated in the original figures and supplementary material (Fig. 2d-f, 2l; Fig. 3f-h; Fig. 4e,f; Fig. 5d,e, 5i,j; Fig. 6g; Supplementary Fig. 4g-i and 4n,o). Given the space limitation, we have removed some of the original panels, including the schematic of the model. We are confident that we have now amply proven the model we had proposed in our original submission.

1. It remains unproven that the changes in TFEB and TFE3 explain the changes in lysosome/autophagy gene expression and function in response to SAHA. Does SAHA still regulate this pathway in TFEB and TFE3 depleted cells or over-expressing cells?

R: To reply to this new comment of the reviewer, we have now tested lysosomal and autophagy gene expression in MEFs dKO for Tfeb and Tfe3 (a kind gift of Rosa Puertollano) that we have further silenced for Mitf. We were able to successfully reduce lysosomal and autophagy gene expression in these cells, even in the presence of HDAC inhibitors. The experimental results are shown below and are presented in Fig. 3f-h, Fig. 5 i,j and Supplementary Fig. 4 g-i.

2. Figures 1 and 2 suggest that TFEB and TFE3 are regulated in a similar way by SAHA. However, Figure 3a and 3b show that the abundance of their transcripts changes in opposite directions following treatment with SAHA.

R. Fig. 1 and 2 show the results of chromatin immunoprecipitation of HDAC2 bound to the promoters of TFEB and TFE3. Instead, Fig. 3a,b show the levels of expression of TFEB and TFE3 (measured by RT-qPCR) in response to SAHA and romidepsin treatments. As the reviewer is certainly aware of, these transcription factors are known to be expressed at different levels in various tissues and cell types. The effect of SAHA and romidepsin on their transcription levels (Fig. 3a,b) likely reflect the relative abundance of the individual transcription factors in these cells. As seen the graphs below, TFE3 is much more abundant in HeLa cells than TFEB (values are normalized to the house keeping gene HPRT1).

3. Does SAHA still regulate the expression of lysosomal genes in cells that lack MYC?

R. As requested by the Reviewer, we have tested the occurrence of lysosomal biogenesis in cells silenced for MYC and treated with SAHA. This experiment was particularly difficult and time consuming because we had to balance the dual effect of silencing MYC and affecting growth rate. Notably, HeLa cells silenced for MYC showed induction of lysosomal genes without the need for additional treatment with SAHA. However, treatment of shMYC HeLa cells with SAHA resulted in the induction of lysosomal gene expression. The experimental results are summarized below and are presented in Fig. 4 e,f; Fig. 6g and Supplementary Fig. 4 n,o.

4. It remains difficult to make a link between changes in gene expression and actual function of lysosomes. For example, NEU1 mRNA levels were found to increase ~16 fold in response to SAHA treatment. Figure S2 suggests that there are similar changes in NEU1 protein levels. Meanwhile, the NEU1 activity only increased ~2 fold (Fig. 1). Can this difference be explained? What happens to NEU1 abundance in lysosomes? Could there be a trafficking defect? The same is true for other lysosomal enzymes where the changes in mRNA levels far exceed changes in their activities. Some speculation on this topic was provided in the response to the reviewers. This should also be more clearly presented in the actual manuscript in order to help readers make sense of this data.

R: The difference between NEU1 mRNA/protein levels and NEU1 enzymatic activity observed upon SAHA treatment could be explained by the low specific activity of NEU1 towards the synthetic substrate and/or by the rate limiting amount of PPCA available for chaperoning and activating NEU1 in lysosomes (Bonten et al 2000). We have amended the manuscript to clarify this point, as suggested by the reviewer.

5. The autophagic flux assays in Figure 4f and g are puzzling as it appears that starvation has no effect on autophagic flux (this goes against the well established induction of autophagy in response to starvation). This raises questions about whether this assay is effectively measuring autophagic flux. A simpler alternative, would be to examine the ratio of lipidated to non-lipidated LC-3 by immunoblot analysis in cells treated +/-SAHA and +/-bafilomycin A. Given that the SAHA part of this experiment was already done in 4b, this is certainly feasible.

R: We have repeated this assay as recommended by the reviewer. The results are shown below and are presented in Fig. 5 d,e.

6. Fig. 1 D. Electron microscopy quantification refers to analysis of 20 images. How many biological replicates does this represent?

R. Two biological replicates were used and this detail has been added in the method section of the manuscript.

7. Why are error bars in the graph presented in 1q presented as SEM while other panels in this figure use SD?

R. To address this comment, and for consistency, we have now presented the data in Fig. 1q with SD instead of SEM.

8. Figure S1 is titled "Lysosomal genes are epigenetically regulated". In reality, all that is shown is that they are regulated by treatment with SAHA.

R. We have now changed the title of this Supplementary Fig. to "Lysosomal genes are regulated by HDAC inhibition".

9. Figure S4 is titled "Supplementary Fig. 4. MYC levels are epigenetically regulated and control the expression of the MiT/TFE transcription factors." In reality, all that is shown is that SAHA treatment has effects on the abundance of MYC, TFEB, TFE3. No causal relationship is established by the data in this figure. In contrast, the titles of Supplemental Figures 5 and 6 are much more reasonable.

R. We have changed the title of this Supplementary Fig. to "MYC levels are regulated by HDAC inhibition and control the expression of the MiT/TFE transcription factors".

Reviewer #2 (Remarks to the Author):

In this revised manuscript, the authors have properly addressed all my critiques. In particular the experiments in which HDAC2 was genetically or pharmacologically disrupted, support the role of this particular HDAC in regulating in regulating MYC.

Overall, this revised version is clearly superior to its predecessor and as such this manuscript should be

accepted for publication.

We are grateful to this reviewer for supporting publication of this work.

Reviewer #3 (Remarks to the Author):

The authors have addressed many of the issues raised by reviewers and the manuscript has been improved substantially. Two major issues remain, as well as a few minor points.

1. The authors conclude that MYC and HDAC interact at the target promoters to repress gene transcription (Fig 9), yet no evidence is provided that MYC and HDACs interact or are even co-bound to the target promoters. Additional evidence, for MYC:HDAC interaction such as proximity ligation assay and/or co-ip are required. In addition, to provide evidence that MYC:HDAC co-regulate target gene transcription ChIP/Re-ChIP showing the MYC ChIP material could then be used to ChIP HDACs would provide experimental support for their model. This is important as this model of MYC:HDAC repression has not been substantiated by others. The authors only cite one BBRC paper from 2004 to support their claim (ref 29).

R. To address the reviewer's concern, we have performed co-immunoprecipitation between MYC and HDAC2 and sequential ChIP on the promoters of TFEB and TFE3 (Fig. 2 d,e and 2 l). In addition, we have now added two more references on this subject. We hope that the reviewer agrees that these additional experiments better support the model we are proposing. For space limitations, we chose to remove the schematic representation of our model.

2. The mechanism of MYC down-regulation in response to HDAC inhibition remains unclear. The authors suggest that HDACs are directly regulating MYC gene transcription and that inhibition of HDACs directly causes the down-regulation of MYC expression. No evidence is provided to show that HDACs are bound to the MYC promoter, per se. Also, one presumes HDACs at the MYC promoter would suppress MYC expression and inhibiting HDACs would lead to the up-regulation of MYC, yet MYC is down-regulated in response to HDAC inhibition. This is a major point of confusion that the authors need to clarify.

R. In response to these comments by the reviewer, we respectfully emphasize that our results are in agreement with those of several investigators who have published that inhibition of HDACs reduces the levels of MYC mRNA and protein (e.g. Lee et al 2003, Li et al 2004, Xu et al 2005, Hideshima et al 2015, Nebbioso et al 2017). In addition, as suggested by the reviewer, we have performed ChIP using HDAC2 antibody on the MYC promoter. The results of this analysis are presented in Fig. 5 f.

It is likely that the down-regulation of MYC is an indirect consequence of HDAC inhibition driving cellular growth arrest/autophagy, which in turn leads to the down-regulation of MYC expression. The authors did not respond to my query regarding the growth state of the cells, which was raised three times in the original review. Perhaps the authors don't understand why this issue is important. The issue is 'cause or consequence'. For example, MYC protein levels decrease in response to shRNAs targeting HDAC2 (Fig.2b). Is this because HDACs regulate MYC transcription, as the authors suggest, or because knockdown of HDACs leads to decreased cell proliferation and therefore MYC is downregulated. Similarly, in response to SAHA treatment, MYC no longer ChIPs to promoters (Fig 2h, i). Is this because SAHA decreases MYC protein levels due to a direct (HDAC regulation of MYC expression; a causal effect) or indirect (SAHA triggered growth arrest, leading to MYC down-regulation; a consequential effect)? This cause or consequence issue repeats throughout the manuscript, e.g. Fig 3a. It would be incorrect to leave the reader thinking that inhibiting HDACs directly down-regulates MYC expression. This needs clarification either through additional experimentation or by including this caveat in the text accordingly.

R. The reviewer is totally right, and we fully understand his/her point. Based on the published literature and our combined results, we would favor a "causal effect" of SAHA treatment on MYC protein levels. However, following the recommendation of the reviewer, we have also remarked on the potential dual effect of HDAC inhibition in the first paragraph of the Discussion of the manuscript, adapting some of the wording of the reviewer, thank you.

Minor issues:

1. Line 152: 'important noticing' should be 'important to notice'

R. We have changed this.

2. Figures: The controls for a given experiment need to be reported as part of that experiment and not as separate figure panels. By displaying as independent panels, there is no confidence that the controls were run at the same time as the experiment. For example, Fig 2b and 2c; supp fig 4a and 4b, etc. These need to be changed, e.g. Fig 2b data (left panel), controls (right panel).

R. For clarity, the control panels belonging to the various Western blots and representing the same membrane stained with Coomassie, are now shown together with the corresponding immunoblots'. This is also detailed in all figure legends.

3. Review article references are very old (from the 1990's) and need to be updated, e.g. Ref 26 and 27

R. Reference 26 (Halazonetis et al) refers to the original paper characterizing the complete binding sequence of MYC (10 nucleotides). In agreement with the reviewer, we have substituted the reference by Lusher et al. 1990 with a more recent review.

REVIEWERS' COMMENTS:

Reviewer #3 (Remarks to the Author):

The authors have been highly responsive to my concerns. More work is warranted in future studies to further interrogate mechanism, as results in MEF dKO +/- Mitf knockdown are not robust, but the bulk of the data presented here are sound, well-controlled and support the concept that, "support the concept that HDACs regulate a molecular switch of gene transcription", as stated in the title.

Reviewer #5 (Remarks to the Author):

Overall:

This manuscript provides ample new evidence for the interaction of Myc with lysosomal and autophagy gene transcription factors TFEB, TFE3, MITF, and the interesting new autophagy TF FOXH1. There is some previous literature connecting Myc to suppression of TFEB (E μ -Myc mice, in a lymphoma system, which has very different characteristics than the current study). However, the previous data are not PubMed indexed and may not be peer reviewed (ref: Blood 2013 122:3784, Blood 2015 126:2450;). The opposite has also been reported; inhibition of Myc leads to autophagosome impairment by JNK signaling (PMID 23933736). This latter study was also in HeLa cells, which is surprising that the results here conclude the opposite. In the HMG paper, only LC3-I or LC3-II individual bands are shown, leading to a suspicious presentation of the results. The HMG paper also relies on p62 expression and punctae, which can aberrantly increase due to redox biology and the unique proteasome / Ub system of HeLa cells, which are HPV+. Conversely, a publication highlighting the role of autophagy in ovarian cancer shows a strong genetic suppression in a disease type with ~80% of tumors overexpressing Myc (PMID 28198375). In addition, autophagy is thought of as a stemness-maintaining pathway, which Myc controls as well. Taken together, the role of Myc in the context of autophagy has little previous literature with an unclear consensus, despite each factor's prime role in almost every cancer type.

The current study contributes to the understanding of the interaction of c-Myc with lysosomal and autophagy genes through direct transcription factor studies and integrates such studies with HDACi therapeutics. The authors do not shy away from complex data and are more transparent with their data than previous publications from other labs (here, detailed box plots, standard deviation, and a broad array of genes and cell line models are used). The data in the current revised form improve on the already strong initial study and are well deserving of the Nature Communications brand.

Specific points regarding reviewer 1 comments:

Reviewer #1 (Remarks to the Author):

This manuscript strings together a wide range of experiments that lead the authors to conclude that MYC suppresses lysosome function and autophagy by interfering with the ability of MIT-TFE and FOXH1 transcription factors to regulate the expression of genes that encode proteins which function in these pathways. This appears to occur both through the regulation of TFEB and TFE3 expression as well as by competing for binding sites in the promoters of target genes and/or recruiting histone deacetylases (HDACs) to such promoters. However, close inspection of the data raises questions about whether the major conclusions are robustly justified. In particular, there is an over-reliance on interpretations that are based on correlations. In particular, although HDAC inhibition has effects

on
MYC, TFEB, TFE3, FOXH1, lysosomes and autophagy, a causal chain of events between them has not been rigorously established. As a result, there is considerable uncertainty about the model proposed in Figure 9 that implies a direct cause and effect relationship. For these reasons, there is a significant mismatch between the confidence with which a novel model is proposed and the actual data provided in support of this model.

R: We have successfully addressed all additional concerns raised by this reviewer. All new data are presented in 19 new panels that we have incorporated in the original figures and supplementary material (Fig. 2d-f, 2l; Fig. 3f-h; Fig. 4e,f; Fig. 5d,e, 5i,j; Fig. 6g; Supplementary Fig. 4g-i and 4n,o). Given the space limitation, we have removed some of the original panels, including the schematic of the model. We are confident that we have now amply proven the model we had proposed in our original submission.

R1A: Overall the authors have addressed the points to a satisfactory level, provided some minor clarification in their labeling in the figures and in their method descriptions which are currently vague. Some of these clarifications are noted in specific responses, below.

Minor comments intended to help get this manuscript publication-ready:

- There is very little evidence that mutation in MiT/TFE factors regulate oncogenesis. Mutations in these genes are rare and follow the background expected rates (cBio, TCGA studies). I would recommend removing "mutation" from the explanation of reference 21, as the expression levels are the only pieces of data with any reliable publication backing.
- Fig S2 reads "significamce" not "significance" in final sentence
- Fig 6f. ULK1 or ULK2 or ULK3?
- Some figures contain asterisk labeled p-values, some don't, even though they are the same type of data
- Fig8. It appears as though two fibroblast isolations per Type I or Type II are plotted, but this is not clear in the figure or legend. There is no loading control for 8b.
- For clarity all ChIP-seq peak data should contain an obvious label for the TSS, as it appears in this diagram the TSS is on the right side of the panel, rather than the left.

1. It remains unproven that the changes in TFEB and TFE3 explain the changes in lysosome/autophagy gene expression and function in response to SAHA. Does SAHA still regulate this pathway in TFEB and TFE3 depleted cells or over-expressing cells?

R: To reply to this new comment of the reviewer, we have now tested lysosomal and autophagy gene expression in MEFs dKO for Tfeb and Tfe3 (a kind gift of Rosa Puertollano) that we have further silenced for Mitf. We were able to successfully reduce lysosomal and autophagy gene expression in these cells, even in the presence of HDAC inhibitors. The experimental results are shown below and are presented in Fig. 3f-h, Fig. 5 i,j and Supplementary Fig. 4 g-i.

R1a: In general, the presented data may be satisfactory to address the previous criticism. However, context for the data is needed.

Please include information for the source of these antibodies in your methods section. If the labels are correct, no WT control lane is presented to validate antibody specificity. However, perhaps the SAHA label is meant to be a dKO label. Please clarify in your S4 legend (panels g,h) these data are from MEFs with knockouts of the target genes (currently, "depleted" – is this shRNA, CRISPR-mediated knockout, TALEN-mediated knockout, or Cre-LoxP knockout? the reference would indicate CRISPR). The legend also notes "Sh C+ sh Control" without explanation or placing in a sentence. For Author Response Figure 3f,g, it is unclear what statistical comparison the asterisks are referring to. If it is to compare dKO shMITF to WT, then an asterisk is expected for MCOLN1 (none is shown). If it is to compare to the null 1.0 expression hypothesis, then the WT may be expected to have asterisks. For figure 3h, no asterisk is shown, although the legend states the dKO shMITF is suppressed relative to WT control. The asterisks in the author response are not shown in the manuscript figure.

The main text describing these figures does not clarify what the data are actually are.

Figures 1 and 2 suggest that TFEB and TFE3 are regulated in a similar way by SAHA. However, Figure 3a and 3b show that the abundance of their transcripts changes in opposite directions following treatment with SAHA.

R. Fig. 1 and 2 show the results of chromatin immunoprecipitation of HDAC2 bound to the promoters of TFEB and TFE3. Instead, Fig. 3a,b show the levels of expression of TFEB and TFE3 (measured by RT-qPCR) in response to SAHA and romidepsin treatments. As the reviewer is certainly aware of, these transcription factors are known to be expressed at different levels in various tissues and cell types. The effect of SAHA and romidepsin on their transcription levels (Fig. 3a,b) likely reflect the relative abundance of the individual transcription factors in these cells. As seen the graphs below, TFE3 is much more abundant in HeLa cells than TFEB (values are normalized to the house keeping gene HPRT1).

R1A: These results are satisfactory. It should not be expected that all related transcription factors are upregulated at the transcript level. MAP4K3, for example, is known to regulate TFEB nuclear localization and subcellular localization may be more informative for function (Nature Communications 2018, Hsu et al). The authors recognize this in Fig 9 as well. However, since the study focuses on transcription and ChIP is performed as a more direct measure of gene regulation than subcellular localization, these data are logical, believable, and necessary to present.

3. Does SAHA still regulate the expression of lysosomal genes in cells that lack MYC?

R. As requested by the Reviewer, we have tested the occurrence of lysosomal biogenesis in cells silenced for MYC and treated with SAHA. This experiment was particularly difficult and time consuming because we had to balance the dual effect of silencing MYC and affecting growth rate. Notably, HeLa cells silenced for MYC showed induction of lysosomal genes without the need for additional treatment with SAHA. However, treatment of shMYC HeLa cells with SAHA resulted in the induction of lysosomal gene expression. The experimental results are summarized below and are presented in Fig. 4 e,f; Fig.

6g and Supplementary Fig. 4 n,o.

R1A: A model for Myc's potential upregulation of lysosomes and autophagy suggests increased protein production via mTORC1 creates an unfolded protein response, due to which autophagy is upregulated in an attempt to compensate for reasons of cellular homeostasis. Since cell cycle is decreased as expected here, these new data provide strong support against the previous model.

4. It remains difficult to make a link between changes in gene expression and actual function of lysosomes. For example, NEU1 mRNA levels were found to increase ~16 fold in response to SAHA treatment. Figure S2 suggests that there are similar changes in NEU1 protein levels. Meanwhile, the

NEU1 activity only increased ~2 fold (Fig. 1). Can this difference be explained? What happens to NEU1 abundance in lysosomes? Could there be a trafficking defect? The same is true for other lysosomal enzymes where the changes in mRNA levels far exceed changes in their activities.

Some

speculation on this topic was provided in the response to the reviewers. This should also be more clearly presented in the actual manuscript in order to help readers make sense of this data.

R: The difference between NEU1 mRNA/protein levels and NEU1 enzymatic activity observed upon SAHA treatment could be explained by the low specific activity of NEU1 towards the synthetic substrate

and/or by the rate limiting amount of PPCA available for chaperoning and activating NEU1 in lysosomes

(Bonten et al 2000). We have amended the manuscript to clarify this point, as suggested by the reviewer.

R1A: The author's response is satisfactory.

5. The autophagic flux assays in Figure 4f and g are puzzling as it appears that starvation has no effect on autophagic flux (this goes against the well established induction of autophagy in response

to starvation). This raises questions about whether this assay is effectively measuring autophagic flux.

As simpler alternative, would be to examine the ratio of lipidated to non-lipidated LC-3 by immunoblot analysis in cells treated +/-SAHA and +/-bafilomycin A. Given that the SAHA part of this

experiment was already done in 4b, this is certainly feasible.

R. We have repeated this assay as recommended by the reviewer. The results are shown below and are presented in Fig. 5 d,e.

R1A. Starvation is not always a reliable inducer of autophagic flux, depending on the method used. HeLa was the first broadly used cell line in part due to its ability to grow regardless of perfect cell culture nutrient content. Please clarify what "starved" means in your Methods section and figure legends. Is it HBSS, a reduction in amino acids, serum reduction, etc?

Please clarify how the LC3B protein levels were quantified (relative to Coomassie? If so, please show). The y-axis labels do not necessarily match their description for all LC3-II blots. Fig 6e says "Quantification of LC3-II levels", which may suggest relative to an actin or other loading control, but the legend states it is relative to LC3-I. Please correct and explain throughout.

As an additional level of confusion for lay readers, the authors do not describe "D" and "S" labels in Figure 5d. Readers are left to guess they are DMSO and SAHA. This is a problem in other figures with even less information in the legend (eg, S4a). Flipping pages to the Methods section, a

sentence reads "For SAHA treatment, cells were incubated for 24 h at 37 °C in medium containing DMSO (Sigma-Aldrich) or 20 μM or when specified 8 μM SAHA (Sigma-Aldrich) diluted in DMSO". This sentence yields further confusion rather than clarification. Please review all figure panels for appropriate description of their labeling.

6. Fig. 1 D. Electron microscopy quantification refers to analysis of 20 images. How many biological replicates does this represent?

R. Two biological replicates were used and this detail has been added in the method section of the manuscript.

R1A: Thank you for the clarification.

7. Why are error bars in the graph presented in 1q presented as SEM while other panels in this figure use SD?

R. To address this comment, and for consistency, we have now presented the data in Fig. 1q with SD

instead of SEM.

R1A: This is a satisfactory response.

8. Figure S1 is titled "Lysosomal genes are epigenetically regulated". In reality, all that is shown is that they are regulated by treatment with SAHA.

R. We have now changed the title of this Supplementary Fig. to "Lysosomal genes are regulated by HDAC inhibition".

R1A: This is a satisfactory response.

9. Figure S4 is titled "Supplementary Fig. 4. MYC levels are epigenetically regulated and control the expression of the MiT/TFE transcription factors." In reality, all that is shown is that SAHA treatment has effects on the abundance of MYC, TFEB, TFE3. No causal relationship is established by the data in this

figure. In contrast, the titles of Supplemental Figures 5 and 6 are much more reasonable.

R. We have changed the title of this Supplementary Fig. to "MYC levels are regulated by HDAC inhibition and control the expression of the MiT/TFE transcription factors".

R1A: This is a satisfactory response.

Signed,
-Joe Delaney

Reply to Reviewers' comments:

Reviewer #3 (Remarks to the Author):

The authors have been highly responsive to my concerns. More work is warranted in future studies to further interrogate mechanism, as results in MEF dKO +/- Mltf knockdown are not robust, but the bulk of the data presented here are sound, well-controlled and, "support the concept that HDACs regulate a molecular switch of gene transcription", as stated in the title.

R: We would like to thank this reviewer for his/her comments and for acknowledging the work we have performed to address his/her concerns.

Reviewer #5 (Remarks to the Author):

Overall:

This manuscript provides ample new evidence for the interaction of Myc with lysosomal and autophagy gene transcription factors TFEB, TFE3, MITF, and the interesting new autophagy TF FOXH1. There is some previous literature connecting Myc to suppression of TFEB (E μ -Myc mice, in a lymphoma system, which has very different characteristics than the current study). However, the previous data are not PubMed indexed and may not be peer reviewed (ref: Blood 2013 122:3784, Blood 2015 126:2450;). The opposite has also been reported; inhibition of Myc leads to autophagosome impairment by JNK signaling (PMID 23933736). This latter study was also in HeLa cells, which is surprising that the results here conclude the opposite. In the HMG paper, only LC3-I or LC3-II individual bands are shown, leading to a suspicious presentation of the results. The HMG paper also relies on p62 expression and punctae, which can aberrantly increase due to redox biology and the unique proteasome / Ub system of HeLa cells, which are HPV+. Conversely, a publication highlighting the role of autophagy in ovarian cancer shows a strong genetic suppression in a disease type with ~80% of tumors overexpressing Myc (PMID 28198375). In addition, autophagy is thought as a stemness-maintaining pathway, which Myc controls as well. Taken together, the role of Myc in the context of autophagy has little previous literature with an unclear consensus, despite each factor's prime role in almost every cancer type.

The current study contributes to the understanding of the interaction of c-Myc with lysosomal and autophagy genes through direct transcription factor studies and integrates such studies with HDACi therapeutics. The authors do not shy away from complex data and are more transparent with their data than previous publications from other labs (here, detailed box plots, standard deviation, and a broad array of genes and cell line models are used). The data in the current revised form improve on the already strong initial study and are well deserving of the Nature Communications brand.

R: We want to thank Dr. Delaney for taking the time to meticulously review the manuscript and our reply to the additional comments of Reviewer 1 that he found satisfactory in most cases. We also thank him for signing his review and for his uplifting comments on the quality of the work. He is setting an example of a collegial, transparent and fair review.

Specific points regarding reviewer 1 comments:

Reviewer #1 (Remarks to the Author):

This manuscript strings together a wide range of experiments that lead the authors to conclude that MYC suppresses lysosome function and autophagy by interfering with the ability of MIT-TFE and FOXH1 transcription factors to regulate the expression of genes that encode proteins which function

in these pathways. This appears to occur both through the regulation of TFEB and TFE3 expression as well as by competing for binding sites in the promoters of target genes and/or recruiting histone deacetylases (HDACs) to such promoters. However, close inspection of the data raises questions about whether the major conclusions are robustly justified. In particular, there is an over-reliance on interpretations that are based on correlations. In particular, although HDAC inhibition has effects on MYC, TFEB, TFE3, FOXH1, lysosomes and autophagy, a causal chain of events between them has not been rigorously established. As a result, there is considerable uncertainty about the model proposed in Figure 9 that implies a direct cause and effect relationship. For these reasons, there is a significant mismatch between the confidence with which a novel model is proposed and the actual data provided in support of this model.

R: We have successfully addressed all additional concerns raised by this reviewer. All new data are presented in 19 new panels that we have incorporated in the original figures and supplementary material (Fig. 2d-f, 2l; Fig. 3f-h; Fig. 4e,f; Fig. 5d,e, 5i,j; Fig. 6g; Supplementary Fig. 4g-i and 4n,o). Given the space limitation, we have removed some of the original panels, including the schematic of the model. We are confident that we have now amply proven the model we had proposed in our original submission.

R1A: Overall the authors have addressed the points to a satisfactory level, provided some minor clarification in their labeling in the figures and in their method descriptions which are currently vague. Some of these clarifications are noted in specific responses, below.

Minor comments intended to help get this manuscript publication-ready:

-There is very little evidence that mutation in MiT/TFE factors regulate oncogenesis. Mutations in these genes are rare and follow the background expected rates (cBio, TCGA studies). I would recommend removing "mutation" from the explanation of reference 21, as the expression levels are the only pieces of data with any reliable publication backing.

R: As suggested by the Reviewer, we have removed the word "mutated".

-Fig S2 reads "significamce" not "significance" in final sentence

R: Thank you, we have corrected the spelling.

-Fig 6f. ULK1 or ULK2 or ULK3?

R: It is ULK1 and has been corrected in the Figure.

-Some figures contain asterisk labeled p-values, some don't, even though they are the same type of data

R: In agreement with the Reviewer, we have added asterisks throughout the panels and the corresponding p values are now reported at the end of each figure legend.

-Fig8. It appears as though two fibroblast isolations per Type I or Type II are plotted, but this is not clear in the figure or legend. There is no loading control for 8b.

R: We have edited the legend and moved the loading control from the Supplementary Figure to Figure 8.

-For clarity all ChIP-seq peak data should contain an obvious label for the TSS, as it appears in this diagram the TSS is on the right side of the panel, rather than the left.

R: We have now marked the TSS in all ChIP-seq panels.

1. It remains unproven that the changes in TFEB and TFE3 explain the changes in lysosome/autophagy gene expression and function in response to SAHA. Does SAHA still regulate this pathway in TFEB and TFE3 depleted cells or over-expressing cells?

R: To reply to this new comment of the reviewer, we have now tested lysosomal and autophagy gene expression in MEFs dKO for Tfeb and Tfe3 (a kind gift of Rosa Puertollano) that we have further silenced for Mitf. We were able to successfully reduce lysosomal and autophagy gene expression in these cells, even in the presence of HDAC inhibitors. The experimental results are shown below and are presented in Fig. 3f-h, Fig. 5 i,j and Supplementary Fig. 4 g-i.

R1A: In general, the presented data may be satisfactory to address the previous criticism. However, context for the data is needed.

Please include information for the source of these antibodies in your methods section. If the labels are correct, no WT control lane is presented to validate antibody specificity. However, perhaps the SAHA label is meant to be a dKO label. Please clarify in your S4 legend (panels g,h) these data are from MEFs with knockouts of the target genes (currently, "depleted" – is this shRNA, CRISPR-mediated knockout, TALEN-mediated knockout, or Cre-LoxP knockout? the reference would indicate CRISPR). The legend also notes "Sh C+ sh Control" without explanation or placing in a sentence. For Author Response Figure 3f,g, it is unclear what statistical comparison the asterisks are referring to. If it is to compare dKO shMITF to WT, then an asterisk is expected for MCOLN1 (none is shown). If it is to compare to the null 1.0 expression hypothesis, then the WT may be expected to have asterisks. For figure 3h, no asterisk is shown, although the legend states the dKO shMITF is suppressed relative to WT control. The asterisks in the author response are not shown in the manuscript figure. The main text describing these figures does not clarify what the data are actually are.

R: We apologize for the confusion and thank the reviewer for identifying mislabeling of the Supplementary Figure S4. The Tfeb and Tfe3 dKO MEFs were generated by Dr. Rosa Puertollano with CRISPR-mediated technology. We have now specified what the acronym ShC stands for. Regarding the statistical analysis, this was performed comparing the normalized expression of lysosomal or autophagy genes in MEFs WT (+ shC) versus dKO (+ shC) [no p values are included because it was not significant], and in MEFs WT (+ shC) versus dKO with silenced Mitf (+ shMitf) [p values are now included for significant comparison].

Figures 1 and 2 suggest that TFEB and TFE3 are regulated in a similar way by SAHA. However, Figure 3a and 3b show that the abundance of their transcripts changes in opposite directions following treatment with SAHA.

R. Fig. 1 and 2 show the results of chromatin immunoprecipitation of HDAC2 bound to the promoters of TFEB and TFE3. Instead, Fig. 3a,b show the levels of expression of TFEB and TFE3 (measured by RT-qPCR) in response to SAHA and romidepsin treatments. As the reviewer is certainly aware of, these transcription factors are known to be expressed at different levels in various tissues and cell types. The effect of SAHA and romidepsin on their transcription levels (Fig. 3a,b) likely reflect the relative abundance of the individual transcription factors in these cells. As seen the graphs below, TFE3 is much more abundant in HeLa cells than TFEB (values are normalized to the house keeping gene HPRT1).

R1A: These results are satisfactory. It should not be expected that all related transcription factors are upregulated at the transcript level. MAP4K3, for example, is known to regulate TFEB nuclear localization and subcellular localization may be more informative for function (Nature Communications 2018, Hsu et al). The authors recognize this in Fig 9 as well. However, since the study

focuses on transcription and ChIP is performed as a more direct measure of gene regulation than subcellular localization, these data are logical, believable, and necessary to present.

R: We thank the Reviewer for his supportive comments

3. Does SAHA still regulate the expression of lysosomal genes in cells that lack MYC?

R. As requested by the Reviewer, we have tested the occurrence of lysosomal biogenesis in cells silenced for MYC and treated with SAHA. This experiment was particularly difficult and time consuming because we had to balance the dual effect of silencing MYC and affecting growth rate. Notably, HeLa cells silenced for MYC showed induction of lysosomal genes without the need for additional treatment with SAHA. However, treatment of shMYC HeLa cells with SAHA resulted in the induction of lysosomal gene expression. The experimental results are summarized below and are presented in Fig. 4 e,f; Fig. 6g and Supplementary Fig. 4 n,o.

R1A: A model for Myc's potential upregulation of lysosomes and autophagy suggests increased protein production via mTORC1 creates an unfolded protein response, due to which autophagy is upregulated in an attempt to compensate for reasons of cellular homeostasis. Since cell cycle is decreased as expected here, these new data provide strong support against the previous model.

R: We again thank the Reviewer for his support of our results.

4. It remains difficult to make a link between changes in gene expression and actual function of lysosomes. For example, NEU1 mRNA levels were found to increase ~16 fold in response to SAHA treatment. Figure S2 suggests that there are similar changes in NEU1 protein levels. Meanwhile, the NEU1 activity only increased ~2 fold (Fig. 1). Can this difference be explained? What happens to NEU1 abundance in lysosomes? Could there be a trafficking defect? The same is true for other lysosomal enzymes where the changes in mRNA levels far exceed changes in their activities. Some speculation on this topic was provided in the response to the reviewers. This should also be more clearly presented in the actual manuscript in order to help readers make sense of this data.

R: The difference between NEU1 mRNA/protein levels and NEU1 enzymatic activity observed upon SAHA treatment could be explained by the low specific activity of NEU1 towards the synthetic substrate and/or by the rate limiting amount of PPCA available for chaperoning and activating NEU1 in lysosomes (Bonten et al 2000). We have amended the manuscript to clarify this point, as suggested by the reviewer.

R1A: The author's response is satisfactory.

5. The autophagic flux assays in Figure 4f and g are puzzling as it appears that starvation has no effect on autophagic flux (this goes against the well-established induction of autophagy in response to starvation). This raises questions about whether this assay is effectively measuring autophagic flux. As simpler alternative, would be to examine the ratio of lipidated to non-lipidated LC-3 by immunoblot analysis in cells treated +/-SAHA and +/-bafilomycin A. Given that the SAHA part of this experiment was already done in 4b, this is certainly feasible.

R. We have repeated this assay as recommended by the reviewer. The results are shown below and are presented in Fig. 5 d,e.

R1A. Starvation is not always a reliable inducer of autophagic flux, depending on the method used. HeLa was the first broadly used cell line in part due to its ability to grow regardless of perfect cell

culture nutrient content. Please clarify what “starved” means in your Methods section and figure legends. Is it HBSS, a reduction in amino acids, serum reduction, etc?

R: We agree with the Reviewer on this point. We have starved the cells using EBSS after extensive washing with HBSS. We have added this detail in the legend and in the Methods.

Please clarify how the LC3B protein levels were quantified (relative to Coomassie? If so, please show). The y-axis labels do not necessarily match their description for all LC3-II blots. Fig 6e says “Quantification of LC3-II levels”, which may suggest relative to an actin or other loading control, but the legend states it is relative to LC3-I. Please correct and explain throughout.

R: We thank the Reviewer for this comment. We have now added to the Figure, the Coomassie-stained membrane corresponding to the immunoblot probed for LC3. We have also specified that the quantifications of the bands on the immunoblots throughout the manuscript were calculated relative to the corresponding Coomassie-stained membranes used as loading control.

As an additional level of confusion for lay readers, the authors do not describe “D” and “S” labels in Figure 5d. Readers are left to guess they are DMSO and SAHA. This is a problem in other figures with even less information in the legend (eg, S4a). Flipping pages to the Methods section, a sentence reads “For SAHA treatment, cells were incubated for 24 h at 37 °C in medium containing DMSO (Sigma-Aldrich) or 20 μM or when specified 8 μM SAHA (Sigma-Aldrich) diluted in DMSO”. This sentence yields further confusion rather than clarification. Please review all figure panels for appropriate description of their labeling.

R: We apologize for the lack of clarity. We have now specified that the D and S abbreviations correspond to D=DMSO, S=SAHA, and amended the Methods section accordingly.

6. Fig. 1 D. Electron microscopy quantification refers to analysis of 20 images. How many biological replicates does this represent?

R. Two biological replicates were used and this detail has been added in the Methods section of the manuscript.

R1A: Thank you for the clarification.

7. Why are error bars in the graph presented in 1q presented as SEM while other panels in this figure use SD?

R. To address this comment, and for consistency, we have now presented the data in Fig. 1q with SD instead of SEM.

R1A: This is a satisfactory response.

8. Figure S1 is titled “Lysosomal genes are epigenetically regulated”. In reality, all that is shown is that they are regulated by treatment with SAHA.

R. We have now changed the title of this Supplementary Fig. to “Lysosomal genes are regulated by HDAC inhibition”.

R1A: This is a satisfactory response.

9. Figure S4 is titled “Supplementary Fig. 4. MYC levels are epigenetically regulated and control the expression of the MIT/TFE transcription factors.” In reality, all that is shown is that SAHA treatment has

effects on the abundance of MYC, TFEB, TFE3. No causal relationship is established by the data in this figure. In contrast, the titles of Supplemental Figures 5 and 6 are much more reasonable.

R. We have changed the title of this Supplementary Fig. to "MYC levels are regulated by HDAC inhibition and control the expression of the MiT/TFE transcription factors".

R1A: This is a satisfactory response.

Signed,

-Joe Delaney